# Hybridization-encoded DNA tags with paper-based readout for anti-forgery raw material tracking

Jiaming Li [1], Alex Crown [2], Peter Ney[1], Sergey Yekhanin[2], Aditi Partap[2], Anuja Shirole[1], Huiting Jiang[2], Sagan Russ [1], Max Gordon [2], Adaora Aroh [2], Jeff Nivala [1], Anirudh Badam[2], Vaishnavi Ranganathan[2], Karin Strauss [2], Ranveer Chandra[2] & Yuan-Jyue Chen [2] ✉

Tracking and tracing raw materials is crucial for securing global supply chains. Conventional methods like barcodes and Radio Frequency Identification (RFID) tags are effective but fall short in ensuring raw material traceability and anti-counterfeiting. This work introduces DNA as a powerful tool for source tracking, leveraging its invisibility, safety, and seamless product integration. We present DNATags–engineered DNA mixtures enabling product labeling with error tolerance–readable in the field via paper tickets that fluoresce under a mobile phone and filter device. Additionally, DNATrack employs DNA Hybridization Encoding (HyEn) for enhanced anti-forgery security. Although current costs are higher ($2-$4 per read and write), declining DNA synthesis costs, along with DNA's unique advantages, make this approach a promising solution for future supply chain management.

Over the past decade, the global supply chain has faced significant disruptions, prompting efforts from organizations such as the United Nations to enhance its robustness and sustainability. In particular, the COVID-19 pandemic underscored the critical importance of building resilient supply chains to meet global demands effectively[1,2]. To counter disruptions, companies are adopting integrated supply chain tracking systems that provide transparency and expedite issue resolution. Traditional tracking methods such as Universal Product Codes (UPC), quick response (QR) codes, radio-frequency-identification (RFID), and near-field communication (NFC) tags, are cost-effective for packaged goods. However, they pose challenges when applied to objects that are very small, flexible, numerous, or physically changing, as well as in scenarios where the tag needs to remain hidden. Conventional tags are susceptible to tampering or replication, particularly impacting high-value items vulnerable to counterfeiting or theft. The removal or replication of tags, like barcodes, complicates tracking and authentication[3].

To address these limitations, the exploration of DNA-based tagging for improved security and traceability in supply chains is gaining traction in both academia and industry[4,5]. DNA possesses inherent characteristics such as its nanoscale size and non-toxic nature[6,7], making it an ideal candidate for integration into various materials. This seamless integration enables the tracking of raw materials and enhances security by rendering removal difficult. Synthetic DNA offers scalability to millions of tags and is cost-effective in terms of reading and writing. The decreasing costs associated with advancements in DNA processing for medical and research applications over the past two decades, coupled with the anticipated continued downward trend[8,9], further bolster the case for adopting DNA-based tagging.

In the realm of supply chain tracking, the ideal design for DNA tags involves scalability, affordability, field adaptability, robustness, and anti-forgery features. Recent studies have made strides in enhancing DNA stability using methods like silica nanoparticles or microbial spore encapsulation[10–13]. Cost-effective reading and writing methods, including nanopore sequencers and paper ticket systems, have also been explored[14,15]. Beyond scientific research, several commercial enterprises have developed DNA-based tagging technologies[4]. For instance, Haelixa employs silica-encapsulated DNA

[1]University of Washington, Paul G. Allen School of Computer Science and Engineering, P.O. Box 1212, Seattle 43017-6221 WA, USA. [2]Microsoft Research, Redmond, WA, USA. ✉e-mail: yuanjc@microsoft.com

tags that can adhere to tangible goods[13]. CypherMark's TraceTag technology uses a pair of primers as detection keys, enabling quantitative PCR-based identification for applications like cash security and anti-counterfeiting[16,17]. Similarly, Tagsmart's Smart DNA Tags integrate synthetic DNA tags with an online platform for artwork certification[18], while Anika Biosciences utilizes bacterial spore-protected microbial tags for food labeling[19]. Despite these innovations, most existing solutions tend to address isolated aspects of DNA tagging rather than offering a comprehensive system design for large-scale functionality. Notably, the widespread replication of DNA has been overlooked, and our research exposes vulnerabilities in existing schemes to forgery through simple laboratory protocols, emphasizing the need for a holistic approach in designing effective tagging systems.

We present DNATrack, a prototype supply chain tracking system featuring DNA tags, aiming at addressing the existing design gap (Fig. 1). DNATrack employs specially designed DNA mixtures, termed DNATags, that store digital information and label products for tracking. We purposefully designed the DNATag to make them inexpensive to manufacture at scale. DNATags incorporate a unique DNA hybridization encoding, termed HyEn, that provides protection against PCR forgery or read-and-rewrite attacks, significantly heightening the tag security compared to existing DNA tagging schemes. We utilized a paper-based readout, termed DNATag reading ticket, that translates DNATags into visual signals representing the stored digital information (Fig. 1 (b)). Users can conveniently read DNATags in the field within 15 minutes at a cost of $2-$4 per read, requiring only basic equipment: an app-equipped smartphone and a fluorescence reader. Our work showcases a full end-to-end workflow, encompassing information encoding methods (both digital and physical), object labeling, to the final field-deployable readout.

In our work, we demonstrated that DNATags reliably induce distinct and correct fluorescence signals on the DNATag reading tickets. We assessed the reaction latency and accuracy. For consistency of the readout, we integrated automated image processing and an error correction code (ECC) in silico. To ensure reliability and reproducibility, we experimented with varied solvent conditions of the DNATags and found out the optimal setup for best stability. We validated DNATrack end-to-end by labeling different objects including lettuce, corks, and petri-dishes with 24-bit DNATags carrying randomly generated barcodes, storing the objects overnight, and reading out the DNATags. Finally, we highlighted the anti-forgery feature of our HyEn encoding scheme by showcasing failed PCR-based forgery against DNATags next to successful forgery attacks of other existing DNA tagging schemes with a simple protocol.

## Results

### Overall workflow of DNATrack

Figures 1a, 2a show the workflow of DNATrack system. Digital information, termed dataword, is encoded with ECC as a codeword. The codeword is converted into DNATags with physical encoding (i.e., HyEn). A multi-bit DNATag is created by mixing various DNA Bits. The DNATag carrying binary signals is then applied to a physical object as a label. To read the information, DNATags are rehydrated and applied to a DNATag reading ticket, inducing a visual pattern in the array of fluorescent dots on the ticket. The visual pattern can be imaged using a smartphone camera and subsequently translated back to the original dataword.

### Hybridization encoding and reaction assessment

A common method for encoding information in DNA involves representing individual digital bits as the presence or absence of pre-specified DNA strands[14,15]. These "DNA bits" are combined to encode arbitrary binary numbers. Berk et al. demonstrated that the presence of a pre-specified DNA strand could generate a fluorescent signal on a paper readout via toehold-mediated strand displacement reactions[15]. While this approach provides a cost-effective way to read DNA tags using paper-based readouts, it suffers from significant vulnerabilities to forgery. For instance, a hacker could amplify a DNA taggant via PCR or sequence the taggant to reproduce it.

To address this security vulnerability, we propose a hybridization-based encoding scheme for representing digital bits. In this method, a

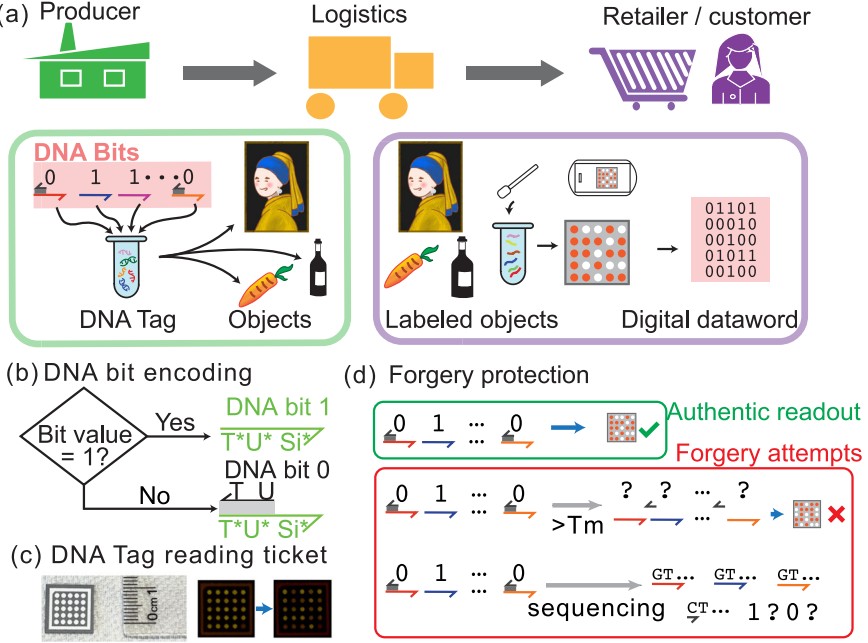

**Fig. 1 | Overview of the DNATrack system. a** Integration of DNATrack in a supply chain. **b** In HyEn, each DNA sequence with a unique Si* domain represents one digit. Bits 1s are represented by single-stranded DNA, and bits 0s by partially double-stranded DNA complexes. **c** Photo and fluorescent images of the DNATag reading ticket. Each spot on the ticket can read one unique digit of the barcode. All spots are at full fluorescent brightness initially. During the readout process after the ticket is exposed to the DNATag, the spots corresponding to digits 1s will turn darker, while the spots corresponding to 0s remain bright. **d** HyEn enables forgery protection.

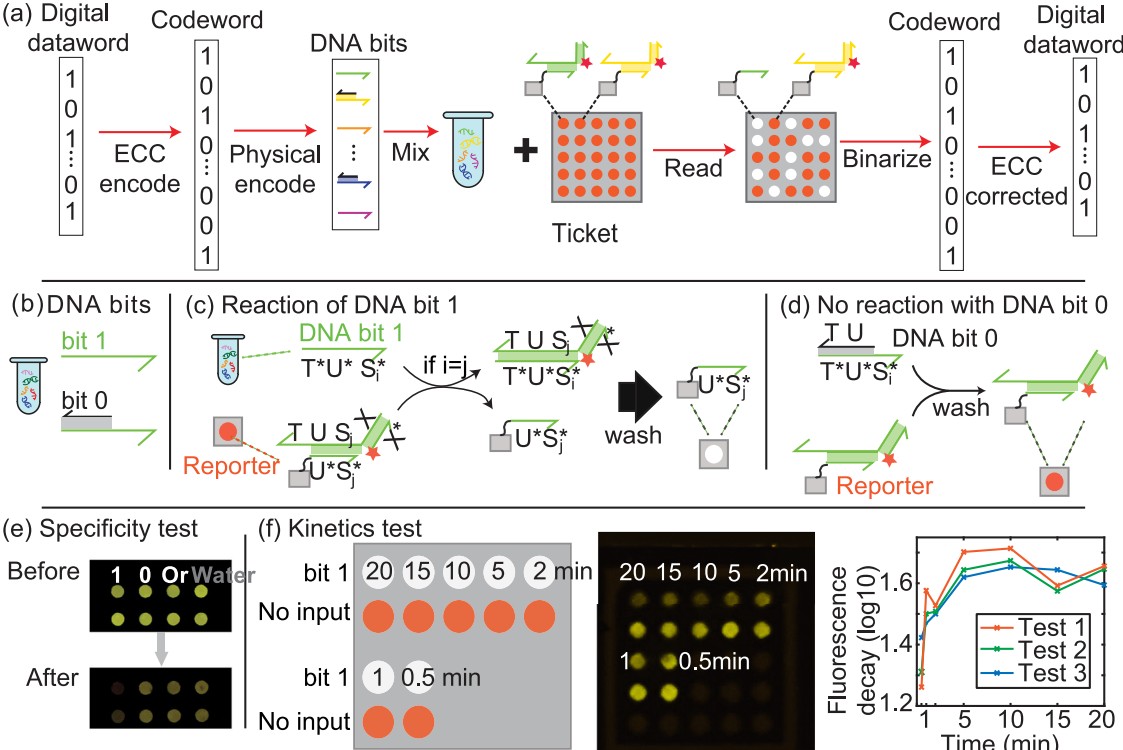

**Fig. 2 | DNATrack design. a** Detailed workflow of encoding and decoding with DNATags. **b** In HyEn, Bits '1's are represented by single-stranded DNA and Bits '0's by partially double-stranded DNA complexes. **c** DNA bit 1 strands can displace the fluorophore-labeled strand from the reporter complex, resulting in a significant drop in fluorescent signals. **d** DNA bit 0 strands cannot initiate the displacement reaction because of the toehold region being blocked. **e** Specificity test of DNA bits. The two rows are replicates for robustness. Only the bit 1 DNA strand resulted in the corresponding reporter spot turning dark. The corresponding DNA bit 0, orthogonal DNA bit 1 that represents another digit in the barcode, and water did not trigger significant fluorescence change in the spots. **f** Kinetics test of the reaction. Three identical tests using different DNA Bits were carried out. The reactions all reached their half life within 5 minutes and saturated around 10 minutes. Source data for (**e**) and (**f**) are provided in the supplemented Source Data file.

*DNA Bit* '1' is single-stranded DNA strand, $< T^*U^*Si^* >$, and a *DNA Bit* '0' is a partially double-stranded DNA complex, where the strand $< T^*U^*Si^* >$ is bound to a blocker strand $< UT >$ (Fig. 1 b). The $T^*$ domain serves as a toehold for initiating strand displacement reaction during the readout, while the $U^*$ domain provides stability of the partial double-strand structure at the intended working temperature. According to our simulation and experimental test, a $U^*$ domain with the length of 15 nucleotides can ensure the stability of DNA Bits at room temperature or below including frozen (Fig. S9). For applications requiring higher ambient temperatures, the $U^*$ domain can be elongated to increase the melting temperature (Tm) (Fig. S16). All DNA Bits share the same sequence in the $T^*U^*$ domains, but each *Bit* carries a unique $Si^*$ domain ($i = 1, 2, 3, \ldots$), corresponding to a distinct digit in the codeword (the 1st, 2nd, 3rd, … digit).

This hybridization encoding method effectively mitigates forgery risks. During a PCR attack, the high-temperature denaturation step dissociates the DNA duplexes representing *DNA Bit* '0's, rendering the encoding irretrievable and preventing recovery of the original information (Fig. 1d). Similarly if an attacker sequences the DNATag, they can determine only the individual sequences used in the DNA Bits, but not the hybridization states encoding the binary digits.

Reading DNATags utilizes toehold mediated strand displacement on a nitrocellulose paper ticket, producing a fluorescent pattern[15,20,21]. This reaction is highly efficient and specific so that the reading can be carried out in a few minutes and natural nucleic acids will not interfere with the reaction. A set of fluorophore-labeled DNA reporter complexes containing different $Si$ domains are arrayed on a nitrocellulose paper ticket (Fig. 2 a). Each reporter complex consists of a probe strand $< XSjUT >$, an anchor strand $< U^*Sj^* >$, and a fluorophore strand $< FX^* >$ (Fig. 2 C). The anchor strand contains a 30-nt poly-T tail on its 3' end for enhanced binding to the nitrocellulose paper (Fig. S3). Upon contact, *DNA Bit* '1' strand $< T^*U^*Si^* >$ can bind the toehold domain of the probe strand and subsequently displace the probe strand together with the fluorophore strand off the anchor strand if the $Si^*$ matches the $Sj$ domain (i.e. $i = j$). The displaced fluorophore strands can be easily washed away, resulting in significantly reduced fluorescent signals. In contrast, the '0's of DNA Bits cannot displace the probe strand because the toehold region $T$ is covered by a blocker strand, resulting in unchanged high fluorescence. Thus, the DNATag reading ticket initially displays an array of uniformly bright fluorescent dots, and it switches to a specific pattern of bright and dim spots upon exposure to a DNATag. We chose the array size to be 5-by-5 in all our experiments, loaded with up to 25 unique reporter complexes decoding 25-bit DNATags. Please see Methods and Supplementary for more details on the fabrication of the tickets. It is noteworthy that our reporter costs approximately half as much as the previous method[15] and even less when scaled up. This is because we utilize only one universal fluorophore-labeled DNA strand for all *DNA Bit* reporters, whereas the other work uses a unique pair of fluorophore- and quencher-labeled DNA strands as a reporter for each single bit.

As a proof of concept, we validated the reaction specificity and measured the kinetics of the on-paper DNA strand displacement reaction using DNA Bits. No leakage was observed when orthogonal (i.e. the $Si^*$ domain of *DNA Bit* does not match the $Sj$ in the reporter complex) *DNA Bit* '1' was applied (Fig. 2(e)). The strand displacement

reaction exceeded half life in the first 5 minutes and reached peak fluorescence decay in 10 minutes (Fig. 2f and S4). We also tested the sensitivity of the reaction with varied concentrations of the reporter complex and DNA Bits (Fig. S2).

## Encoding DNATags with error correction code

Writing and reading DNATags can introduce errors due to factors like DNA synthesis and inherent imprecision of pipetting. To ensure accurate and robust decoding, we employ error correction coding (ECC), which are techniques for detecting and correcting errors in data. Our approach uses a linear code, which minimizes redundant bits while still allowing error correction in short messages[22-24].

In this process, we start with the binary dataword (the original information we want to encode) and multiply it by a generator matrix. This transformation yields a longer codeword, which includes both the original data and additional redundant bits specifically designed to detect and correct errors. These redundant bits provide tolerance for up to a few errors in the codeword. The codeword is then directly encoded by DNATag and when it is read by the ticket, our ECC-integrated decoder uses these redundant bits to retrieve the original dataword, correcting any errors that may have occurred (Fig. 2a). The amount of redundancy (number of redundant bits) can be adjusted based on the application's error tolerance and expected error rate. In our specific demonstration, we chose a 12-bit dataword and a 24-bit codeword (Fig. 3d), giving us the ability to correct errors of up to 3 bits in the codeword readout[22].

## Imaging processing and decoding

To ensure consistency for readout of multi-bit DNATags, we developed an automatic decoding process that includes two stages: image processing and ECC decoding (Fig. 3). First, it converts fluorescence images of the reading ticket into a binary string using an image processing pipeline (Fig. 3(a)). Second, the binary string is decoded to the original information using ECC (Fig. 3(d)).

In the first stage, two greyscaled fluorescence images from the reporter ticket – one pre-reaction and one post-reaction – were fed into the image processing pipeline. The fluorescence spots were masked using Blob detection, and the mean intensities of each spot were calculated. We created a matrix by subtracting the fluorescence intensities of each spot before and after the reaction, and used it as an analog output to quantify the fluorescence decay resulted from the DNATag. The matrix was then normalized and each spot was binarized to 0 or 1 using a universal hard-cut threshold. To determine the value of the threshold, we conducted multiple experiments testing random 24- or 25-bit datawords encoded with the DNATags and measured the fluorescence decays (Fig. 3(c)). We set the threshold value to be 0.38, resulted in the fewest false binarized digit.

In the second stage, the decoder processes the binary string (codeword) to recover the original dataword by iterating through all possible datawords and selecting the dataword whose generated codeword has the smallest Hamming distance to the binary string obtained from the previous stage. This brute-force nearest neighbor decoding is guaranteed to have correct decoding when the error is less than a pre-determined threshold.

## Reliability and end-to-end testing of DNATrack

We demonstrated the robustness of the end-to-end DNATrack system on different object surfaces including petri-dishes, corks and lettuce (Fig. 4 (a)). A random 12-bit ID was digitized as the dataword, encoded by ECC into a 24-bit codeword, and converted to DNATags using HyEn. The DNATag was dried on the labeled object for tagging. After storage overnight, the DNATag was rehydrated and read out using a ticket followed by the automated decoding pipeline (Fig. 4 (b, c)). All nine DNATags applied to the three object surfaces were successfully retrieved and correctly decoded. A video showcasing the end-to-end testing is available in the supplementary material.

One crucial step to ensure that DNATags stay stable during the drying process is to have trehalose mixed in the DNATag solution. We collected data through excessive experiments and compared them using the two-sample $t$-text. We validated that without trehalose, the DNATags after dried and rehydrated will generate more errors than

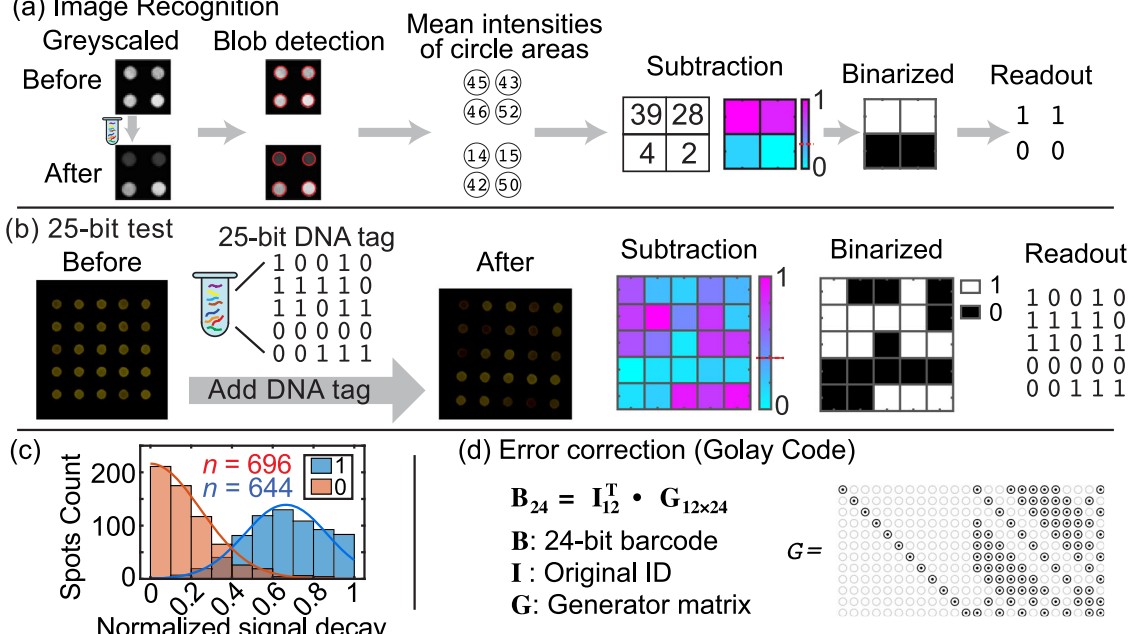

**Fig. 3 | Automated image processing and ECC. a** Automated image processing pipeline to translate fluorescent images into quantified fluorescence data and digitized readout. **b** Example of the pipeline dealing with the readout of a 25-bit DNATag. **c** Statistical summary of the normalized fluorescence decay from 696 DNA Bit '1's and 644 DNA Bit '0's. The bar graphs are the collected experimental data, and the curves are fitted by Gaussian model. More details about the fitting can be found in Methods section. The sample sizes are marked as '$n$'. **d** Error correction code (ECC). Source data for (**c**) are provided in the supplemented Source Data file.

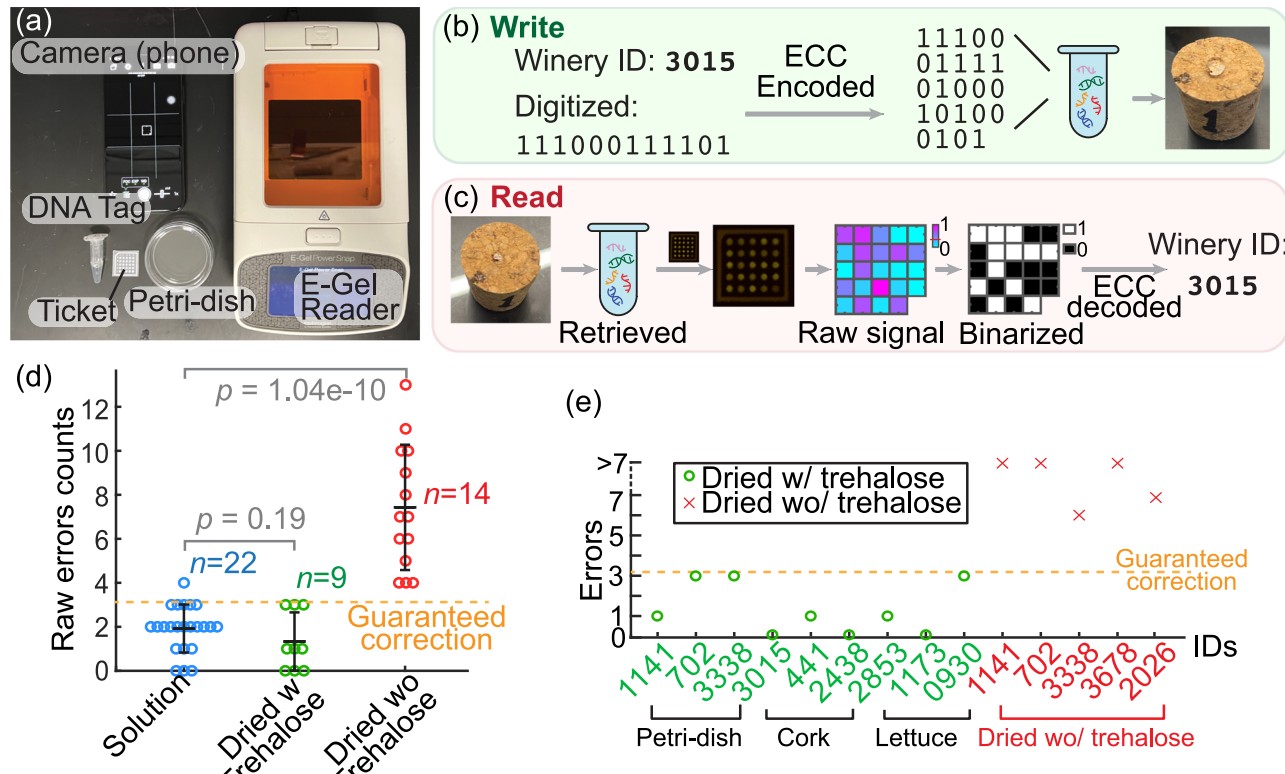

**Fig. 4 | End-to-end applications of DNATrack. a** Photo of essential components used for writing and reading the DNATag. **b** Procedure of encoding a winery ID with DNATag and attaching it to a cork. **c** Exemplary result of retrieving the DNATag from the cork and decoding it with the reading ticket. **d** Comparison of raw errors from reading tickets under different application conditions. Each data point represents the ticket readout of a unique randomly generated DNATag. The sample sizes are marked as '$n$'. The error bars present mean values +/- standard deviation. The $p$ values were calculated using the two-sample $t$-test. **e** Examples of random IDs tested end-to-end on different objects and their raw errors. All DNATags with trehalose were correctly decoded with ECC. Source data for (**b**) to (**e**) are provided in the supplemented Source Data file.

fresh DNATag solutions at 5% significance level and exceed the ECC tolerance (Fig. 4 (e)). Whereas with trehalose, their readout accuracy was the same as fresh DNATag solutions without drying (Fig. 4d, S7). The hypothesized reason is that the double-stranded DNA bits '0' would denature during drying since the hydrogen bonds that maintain the double-strand structure highly rely on the aqueous environment. Trehalose with adequate hydroxyl groups can serve the same role as water even after dried and thus helps maintain the double-stranded structures of the DNA bits[25,26].

To better understand the long-term stability of DNATags, we focus on two primary factors: DNA strand damage and bit-exchange leakage. DNA strand stability has been extensively studied in DNA storage research[27,28]. Organick et al. reported that DNA molecules are least stable in solution, with a half-life of approximately one year. However, this half-life can be extended to over 10 years when stored in trehalose and may exceed 100 years under 'dry' conditions, depending on the storage methods used[27].

Beyond strand damage, bit-exchange leakage is another key consideration for DNATag stability. Since our DNATags encode information in metastable states (i.e., partially double-stranded and single-stranded DNA), Bit '1's and Bit '0's gradually rehybridize to reach their equilibrium state over time. The pathways for bit-exchange are illustrated in Fig. S15a. Leveraging the extensively studied thermo-dynamics of DNA hybridization in solution[29,30], we simulated these exchange pathways and found that DNATags remain stable with minimal kinetic leakage in solution for over 10 years (Fig. S15b). Under dry conditions, we tested the stability of dried DNATags for up to 15 days and all DNATags stayed stable after the storage (Fig. S8). For longer term, we expect the bit-exchange leakage to be lower than in

solution due to the significantly reduced mobility of DNA molecules at dry state.

## Authentication and forgery protection
Tagging not only aids in supply chain tracking but also enhances security by authenticating items and deterring counterfeiting. A robust tagging system should be difficult to forge, as easy duplication or transfer of tags could enable counterfeiters to misuse them, potentially masking the origin of stolen goods to falsify provenance.

Since all DNA tags are physically present on a tagged material, it is difficult to make any tagging scheme completely forgery resistant to a motivated and capable adversary. For example, all DNA tagging schemes can be forged if the tagging material can be moved from a legitimate tagged product to a counterfeit one (e.g., by swabbing DNA). However, we believe it is valuable for DNA tags to be resilient against basic attacks that can be scaled up to forge many items.

In this section, we show how the unique strand hybridization encoding HyEn we use to generate DNATags makes two kinds of forgery more difficult to do at scale. For this discussion, we assume that the adversary has access to some DNA tagged product that they want to counterfeit.

### I. Tag Amplification Attacks
If an adversary is capable of chemically or enzymatically amplifying a DNA tag then they can take a small sample from a legitimate DNA tag and increase it's yield. The higher yield product acts like a high con-centration tag that can be diluted and spread over as many objects as desired. We call this a tag amplification attack.

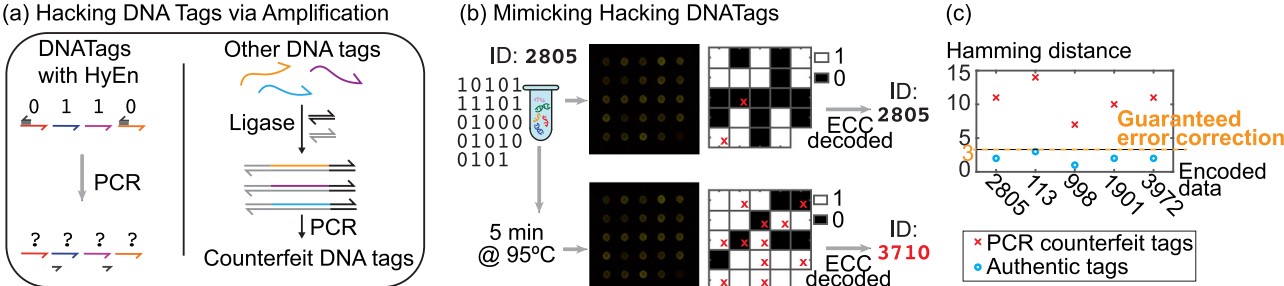

**Fig. 5 | HyEn protects DNATags from forgery attacks. a** Any amplification attack will break apart the double stranded complexes and randomize the tag data. On the other hand, other DNA tags encoded with moderately sized, linear sequences can be enriched easily without knowing the sequences. **b** After incubating the DNATag at 95 °C for 5 minutes, the tag data got completely randomized. The red crosses mark the incorrectly identified bits. **c** Five different DNATags were tested with our without going through the simulated amplification forgery attack. Without the attack, all were decoded with Hamming distances below 3 between the codeword and the ticket's binary output, guaranteeing that ECC can correctly decode them. After the attack, all Hamming distances went over 3 so ECC will generate false readout. Source data for (**b**) and (**c**) are provided in the supplemented Source Data file.

Using basic molecular techniques, like PCR, most tagging schemes can be forged without the knowledge of the tagging sequence. DNA tags composed of moderately sized, linear sequences can be concatenated with strands of known sequence and amplified using PCR; the known sequences do not affect reading and decoding. This protocol can be adapted to work with either single or double stranded DNA. We validated this approach by successfully forging tags from[31] (Fig. S12). Note that this amplification attack will not work with tagging schemes that rely on circular DNA or have tags include in a larger genome like[11].

In the case of HyEn encoded DNATags, any amplification technique that cycles temperature, like PCR, will break apart the double stranded complexes, which effectively randomizes the tag data (Fig. 5 (b) and (c)). To experimentally validate the anti-counterfeiting capability of DNATag against a PCR amplification attack, we incubating the DNATags at 95 °C for 5 minutes after preparation (Fig. 5). This heating step is necessary for all PCR reactions. The DNATags were then cooled down to room temperature and applied to the tickets. We used the same 5 DNATags and ECC. After the mimicked attack, all 5 DNATags generated random patterns in their readouts and decoded into different barcodes.

## II. Read-and-Rewrite Attacks

Conceptually the read-and-rewrite attack is simple; the adversary 'reads' (sequences) the DNA molecules in the tag and then 'rewrites' (synthesizes) them using the same methodology that is used to write legitimate tags. In effect, once the raw tag information has been read, the adversary generates tags like the normal tagging service.

This is a challenging attack to prevent entirely, but it can be made practically more difficult if tags cannot be read via DNA sequencers. All other tagging schemes we are aware of can be read directly with DNA sequencers[11,14,15,31–35]. However, the DNA hybridization encoding we created (HyEn) cannot be sequenced directly. The reason is that both 0 and 1 DNA Bits contain the same DNA strand and what distinguishes them is the presence or absence of a partially double stranded complex. In effect, a forger would sequence the same strands for all DNATags, regardless of what tag was encoded. If the attacker had access to the reading ticket then they could determine which bits were single or double stranded and write them accordingly. However, in security sensitive applications, it is reasonable to restrict access to reading tickets.

A more sophisticated hacker could perform an advanced read-and-rewrite attack. In this scenario, the DNA is extracted from the product and split into two fractions. The first fraction is sequenced to identify all sequences in the taggant, and reporter oligonucleotides are synthesized to create a "self-made" ticket. This ticket is then used to test the second fraction of DNA, identifying which sequences in the taggant are blocked and thereby revealing the codeword.

This attack can be mitigated using several approaches. First, DNA Bits can be chemically modified at their ends to render them incompatible with sequencing, as sequencing typically requires ligation of DNA to sequencing adapters. Another approach is to introduce redundant DNA sequences that share the same structural format as DNA Bits (i.e., $< T^*U15^*S22^* >$). A large pool of random redundant sequences can be inexpensively generated by incorporating random mixed bases in the $Si$ domain (e.g., NNNN..., where N represents a mixture of A/T/C/G). This strategy yields $4^{20}$ possible sequences for the 20-nt $Si$ domain, making it prohibitively expensive to reverse-engineer the authentic sequence from this vast set of random sequences.

## Discussion

In summary, we introduce a supply chain tracking system, DNATrack, which utilizes DNATags to label and track individual products. These DNATags incorporate unique physical encoding, making forgery attempts significantly challenging, particularly against PCR amplification and read-and-rewrite attacks. The system, supported by a mobile application and an eco-friendly paper-based visual readout, allows for convenient field reading of DNATags at a low cost per use. Through rigorous experimentation, DNATrack demonstrates the effectiveness of its encoding scheme, showcasing resistance against forgery attempts. Additionally, efforts to improve DNATag stability, such as using trehalose, are highlighted, further bolstering the system's reliability in real-world applications.

To assess the practicality of this technology, we provide a summarized comparison of existing DNA tag systems (Table 1) and a detailed cost analysis of DNATrack (Table 2). To be viable for supply chain applications, cost, scalability and field usability are of primary concern. Molecular tags from Doroschak et al.[14], which utilized nanopore readers, are presently the fastest (1 minute) for reading. However, their cost per use is approximately $11, making them about four times more expensive than DNA tags using paper-based readout (i.e., ours and Berk et al.[15]). While it is challenging to write large quantities of data using DNATags, this method is well-suited for encoding smaller amounts of information ( <100 bits), ideal for supply chain applications. Our work also significantly enhances security features compared to existing technologies. In summary, DNATrack offers the best fit for global supply chain applications, with sufficient multi-bit capability, low cost, relatively fast turnaround, and advanced security features, despite difficulties in writing large quantities of data in a single mixture. Additionally, although our demonstrations used a gel reader coupled with a smartphone to acquire fluorescence images, the gel reader can be easily replaced with an inexpensive (under $50) device consisting of blue and UV LEDs and an orange filter (Figure S14). This

**Table 1 | Comparison between existing implementations of DNA tagging schemes**

| Work | Writing method | Reading method | Cost per use | # of bits demonstrated | Anti- read-and-rewrite attacks | Anti-amplification attacks | Turnaround time |
|---|---|---|---|---|---|---|---|
| This work | DNA HyEn | Paper ticket (visual) | $2.77 | 25 | Yes | Yes | 10-30 mins |
| 15 | ssDNA | Paper ticket (visual) | $2.72 | 3 | No | No | 10-30 mins |
| 13,31–35,45,46 | ssDNA | PCR, qPCR | N/A | 1-5 | No | No | 1-5 hours |
| 11 | Microbial DNA | PCR, Sanger sequencing | N/A | 22 | No | No | 1 hour |
| 14 | dsDNA | Nanopore sequencing | $11 | 96 | No | No | 1 min |

**Table 2 | Cost of DNATags per write-and-read**

| Components | DNA amount (pmole) of 1 bit | Cost of 1 bit | DNA amount (pmole) of 25 bit | Cost of 25 bits |
|---|---|---|---|---|
| DNA taggant | 240 | $0.1 | 6000 | $2.5 |
| DNA reporter | 0.5 | $0.0009 | 12.5 | $0.0225 |
| Nitrocellulose paper | | $0.25 | | $0.25 |
| Buffer and ink | | $0.0015 | | $0.0015 |
| Total | | $0.3524 | | $2.774 |

The DNA cost shown is based on the manufacturer, IDT.

device can be powered by a low voltage battery, making DNATrack field-deployable.

Although the current cost of DNATag remains high ($2.77 for a 24-bit DNATag), which may present challenges for adoption in low-margin markets like agriculture, a promising entry point for DNATag is in enhancing the anti-counterfeiting measures of luxury goods—such as wines, paintings, and jewelry—that are particularly vulnerable to counterfeiting due to their high value and profit margins. These products often rely on existing anti-counterfeiting technologies, such as RFID, NFC chips and hologram tags. DNATags, with their molecular-level security features, offer a unique layer of protection that could further safeguard these high-value items.

Despite its advantages, DNATrack has limitations. One major challenge is the significant DNA quantity required for a single readout (~60 μg per destructive readout), primarily due to the large reaction buffer (60 μL) needed to cover the DNATag-reading ticket (~1 cm²). Since reaction speed is semi-proportional to DNA concentration, the large buffer necessitates higher DNA inputs to ensure efficient detection. To address this, smaller DNATag-reading tickets with reduced size of reporter spots can be produced using a microarray printer, or the reaction time can be extended, as signal intensity is semi-proportional to both DNA concentration and reaction time.

Future developments aim to expand DNATags' compatibility with various objects. Current DNATags are designed for attachment to solid surfaces but may be easily washed away. Research into waterproof coatings could prevent damage from washing or abrasion while maintaining rehydration ease for readout. Moreover, attaching DNA-Tags to silica or magnetic nanoparticles and encapsulating them in silica could extend their utility, enabling applications in labeling fabrics, liquids, and other diverse objects[27,32,36,37]. Meanwhile, faster turnaround time will also be our future goal. We plan to achieve it by utilizing multiple sets of toehold domains $T$ to minimize the kinetic crosstalk between different DNA Bits and selecting an optimal fluorophore that yields a higher signal-to-noise ratio. Lastly, while DNATags improve security, further investigation is needed to address potential forgery attacks. With the increasing accessibility of DNA amplification and sequencing, identifying and mitigating vulnerabilities is crucial to ensure DNATrack's long-term effectiveness.

## Methods
### Designing highly orthogonal DNA sequences
To minimize unintended interactions, we designed the barcode domains $< Si >$ to be orthogonal using the following in silico process. Random sequences at a length of 20 nucleotides (nt) were generated using a three-letter code (A, C, and T) to reduce secondary structures, with 30% to 70% GC-content (note: now there is only 'C' in the sequence that makes up the 'GC-content') to ensure similar melting temperatures; no more than 4 A's or T's were allowed in a row, and no more than 3 C's were allowed in a row to reduce synthesis errors; no more than continuous 10-nt sequences were allowed to be matched in the sequence library. Any DNA strand showing basepairing interactions by itself (called secondary structures) was filtered out because it would inhibit strand displacement reactions. DNA structures were predicted using the NUPACK design module[38,39]. To increase the stringency of sequence orthogonality, we used the basic sequence alignment program BLAST to screen out domains with long stretches of similar sequences[40]. We obtained 127 orthogonal sequences starting from 10,000 random sequences. The number of orthogonal sequences can be scaled up by using the enormous sequence diversity of 20-bp sequences ($4^{20}$).

### Fabrication of the DNATag reading ticket
Nitrocellulose (NC) paper (Cytiva Life Sciences 1060001) was used as the substrate for the reporter ticket because it easily binds DNA molecules[15,41,42]. NC paper has different binding affinities to molecules with different components or configurations, making it easy to improve the signal/noise ratio by specifically strengthening the binding of DNA that produces the signal and weakening the binding resulting in noise.

We referred to the paper ticket fabrication protocol established by Lu et al.[43]. We first printed the designed ticket patterns onto the NC paper sheet using a wax printer (Xerox Colorqube 8580) (Fig. S1). We baked the printed NC paper at 90 °C for 10 minutes, during which the wax melts and penetrates through the paper. The wax is very hydrophobic and thus creates wells in the NC paper, which separates different reporters. We then cut the paper into individual tickets and stored them at room temperature until the next step (Fig. S1).

### Loading DNATag reporter assays to the reading ticket

All DNA molecules in this study were purchased from Integrated DNA Technologies (IDT). Each bit of DNA reporters consists of three oligonucleotide strands including $< FX^* >$, $< 30TU^*Si^* >$, and $< XSiUT >$ (Fig. 2(c), Fig. S1 (b)). To assemble the reporter complex, we mixed the oligonucleotide strands at a final concentration of $2\,\mu M$ of $< FX^* >$, 2.2 $\mu M$ of $< 30TU^*Si^* >$, and $2.4\,\mu M$ of $< XSiUT >$ in 1x Tris-EDTA (TE) buffer (Corning 46-009-CM) with 10 mM $MgCl_2$ (Sigma-Aldrich 7786-30-3). The mixture went through an annealing protocol starting with a 95 °C incubation for 5 minutes, followed by a gradual temperature decrease at a constant speed of 1°C/minute down to room temperature (Fig. S1(b)). Then, the reporter complex was diluted to $0.2\,\mu M$ with 1x TE buffer with 10mM $MgCl_2$ and spotted onto the wells of the previously prepared paper tickets, $0.3\,\mu L$ per spot. We let it dry at room temperature for at least 20 minutes, washed the ticket by gently flipping it over several times in water to get rid of excessive molecules, and then the reporter ticket was ready to use.

### Collecting visual data from the reading ticket via fluoresence imaging

To collect a visual readout from the reporter ticket, we put the ticket in the E-Gel Electrophoresis Reader (ThermoFisher G8100), turned on the blue light, and took a photo through the orange filter window with a smartphone (Apple iPhone 12 with the camera app 'NightCap') (Fig. 4(a)). We used a constant set of camera parameters for consistency of the images: ISO = 500, exposure = 1/100 s, and white balance = 4000.

### Preparation of individual DNA Bits

DNA Bit '1's are single stranded oligos $< T^*U^*Si^* >$ directly diluted to $100\,\mu M$ from stock using 1x TE buffer with 10 mM $MgCl_2$. Each DNA Bit '0' consists of a tag strand $< T^*U^*Si^* >$ and the universal blocker strand $< UT >$. We first mixed the two strands at final concentrations of 100 $\mu M$ of $< T^*U^*Si^* >$ and $115\,\mu M$ of $< UT >$ in 1x TE buffer with 10 mM $MgCl_2$. The mixture then went through the same annealing protocol as the reporter complex assembly and was ready to use. Both DNA Bits '1's and '0's were stored at -20 °C until use.

### Specificity test of DNA Bits

A 1-bit reading ticket was prepared by loading the same reporter complex (carrying the same $< Si >$) to 8 spots on one ticket. To conduct the test, the corresponding DNA Bit '1', corresponding DNA Bit '0', orthogonal DNA Bit '1', and water as a negative control were separately dropped onto individual reporter spots at the volume of $1\,\mu L$ per spit. After incubation of 20 minutes, the ticket was rinsed in water, dried, and the fluorescence image of the ticket was taken as the result.

### Kinetics test of DNA Bit readout reaction

A 1-bit reading ticket was prepared by loading the same reporter complex (carrying the same $< Si >$) to multiple spots on one ticket. A fluorescence image of the ticket was taken. At each varied time point, 1 $\mu L$ of the corresponding DNA Bit '1' was pipetted onto one individual reporter spot, and water was pipetted onto another reporter spot at the same time for comparison. After 20 minutes from applying the first drop, the ticket was rinsed in water to remove DNA Bits and terminate all readout reactions at the same time. The ticket was then dried and a second fluorescence image was taken. The fluorescence intensity decays of each spot was calculated by subtracting the two images before and after reaction, and was plotted as a measurement of the DNA Bit readout reaction kinetics.

### Encoding digital IDs with DNATags

The digital ID was first binarized to the 12-bit dataword, which was then processed through ECC generating the 24-bit codeword. We assigned every digit of the binary codeword to one DNA Bit with a unique

barcode domain $< Si^* >$. To encode the codeword, we simply mixed all DNA bits representing the '1's and '0's from the codeword at equal molecular amount. We diluted the final mixture to $4\,\mu M$ per DNA bit with 1x TE buffer containing 10mM $MgCl_2$ and 0.7M Trehalose (Research Products International 6138-23-4), and 60 $\mu L$ of such mixture serves as one DNATag. It can then be either dropped and dried on the object surface to label the specific object or applied to the corresponding reporter ticket for decoding.

### Reading the DNATag solution with the reading ticket

We initially took a fluorescence image of the un-used DNATag reading ticket. We next dropped the 60 $\mu L$ DNATag solution to the surface of a clean Petri dish and covered the droplet with the reporter ticket upside-down making sure that all reporter spots were soaked and wet. We then left it at room temperature for 20 minutes, washed the ticket, let it dry, and took the second image for readout.

### Labeling objects with DNATags and retrieving the DNATags for readout

We labeled the corks, lettuce, or petri-dishes with DNATags by dropping the 60 $\mu L$ DNATag solution onto the surface of the object and letting it dry. The labeled object was kept at room temperature overnight before readout if not specified otherwise. To retrieve the DNATag, we dropped 60 $\mu L$ water onto the dried DNATag, pipetted up and down several times to dissolve the DNA, and move the solution to a tube as the retrieved DNATag solution. The solution can then be applied to the DNATag reading ticket for readout.

### Gaussian fitting of normalized signal decays

We gathered the normalized fluorescence decays of all reporter spots from all our well-controlled experiments (i.e., excluding those experiments aimed at testing varied experimental conditions) and separated the data into two sets by whether they encountered corresponding Bit '1' or '0'. We directly fitted the dataset of Bit '1' using truncated Gaussian distribution, setting the truncation boundaries as '0' and '1'. For the Bit '0' dataset, since the expected mean value is 0 because there should be no reaction, while the actual data were all larger than 0, we created a dummy dataset by pooling together the Bit '0' dataset and their opposite values, and fitted this dummy dataset with Gaussian distribution.

### Reporting summary

Further information on research design is available in the Nature Portfolio Reporting Summary linked to this article.

## Data availability

All source data, including sheets of presented data in graphs and uncropped fluorescent images of the reading tickets, are available on GitHub at the following https://github.com/microsoft/DNATagging/or on Zenodo via https://doi.org/10.5281/zenodo.14933948[44]. Source data are provided with this paper.

## Code availability

The code used to analyze the fluorescence images, process the signal, and generate the results in this study is publicly available and has been deposited on GitHub at https://github.com/microsoft/DNATagging/ and in Zenodo at https://doi.org/10.5281/zenodo.14933948, both containing the identical version under the MIT license[44].

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

## Acknowledgements

We would like to thank C. Thachuk from University of Washington for offering input to our paper. This work was supported by a sponsored research agreement (gift funding) by Microsoft.

## Author contributions

J.L. and S.R. performed experiments. A.C. and H.J. developed the image analysis algorithm. A.P., A.C., and S. Y. designed the error correction code. J.L., A.C., and Y.J.C. designed benchmarks, analyzed data and wrote the manuscript. P.N. performed security analysis and contributed to drafting the manuscript. A.A. and M.G. explored experimental protocols and encoding methods. P.N., S.Y., A.P., A.S., J.N., A.B., V.R., K.S., and R.C. contributed to manuscript revisions. K.S., R.C., Y.J.C supervised the work.

## Competing interests

A.C., S.Y., A.P., H.J., M.G., A.A., A.B., V.R., K.S., R.C., and Y.J.C are or were employees at Microsoft Research. Others have no competing interests.
