## [Transparent Peer Review File · Nature Communications]

Hybridization-Encoded DNA Tags with Paper-Based Readout for Anti-Forgery Raw Material Tracking

Corresponding Author: Dr Yuan-Jyue Chen

Version 0:

Reviewer comments:

Reviewer #1

(Remarks to the Author)

The manuscript by Li et al. presented a very interesting DNA hybridization-based method for encoding and tracking in the food supply chain. Compared to the standard labels on the outer packaging, DNA oligonucleotides can be directly applied to the surface of the raw material, thus are more difficult to remove or change. In the DNA oligonucleotide "tags", each digit was represented by DNA hybridization status (single-stranded = 1, double-stranded = 0) instead of by base sequences. Therefore, the encoded information can be easily erased by heating, and difficult to replicate or decode by PCR or sequencing. The authors developed an integrated system for tracking, and tested the feasibility on different types of surfaces. The cost for labeling and readout was reasonable, and the process was not too complex. This is an inspiring application of DNA-based information storage and DNA hybridization.

The manuscript is well written, and the figures are clearly presented. I have a few questions for the authors, and additional experimental data might be needed to enhance the depth and clarity of this work:

1. In non-cold chain transportation, the tagged material might experience medium-high temperature in tropical/subtropical areas, and even in temperate zone during summer. Will the encoded information be erased at 30-60°C during shipping? Can this method be applied in cold chain transportation, especially in frozen status? Please discuss the temperature tolerance of the hybridization-based DNA tags.
2. The DNA tags might be applied on dry or wet surfaces. In dry conditions, how long can the double-stranded DNA last without losing its secondary structure? In wet conditions, since the single-stranded and double-stranded DNA probes seem to have the same sequence in "T" and "U" domains in Figure 2, the "TU" oligos may slowly shift to the single strands after long storage or transportation, making it difficult to distinguish 0 and 1 signals. How long can the tags work properly?
3. Will the DNA tags be sabotaged or contaminated by rubbing or washing?
4. Since plant or animal-derived raw material have intrinsic nucleic acids, will these DNA or RNA affect the readout?
5. The limitations of this tagging method need to be discussed in detail, including types of compatible material, required shipping and storage conditions, etc.
6. This method currently requires 10-30 min turnaround time. Is it possible to speed it up by optimizing the readout reactions and processes?

Minor points:

1. Description of the error correction code is a bit hard to follow. Additional description is needed.
2. There are a few typos in the manuscript, such as "This work proposes the use DNA, with..." in the Abstract.
3. Please check the reference numbers, such as "Nitrocellulose (NC) paper was used as the substrate for the reporter ticket because it easily binds DNA molecules [?, 23, 33]." in the Methods section.

(Remarks on code availability)

Reviewer #2

(Remarks to the Author)

The publication describes a new system for labeling physical objects called DNATrack employs specially designed DNA mixtures, termed DNATags (made of single-stranded and partially double-stranded DNA), for product tracking. The authors described in detail the experiments performed, as well as compared the cost of the proposed technology with others described in the scientific literature. I think that the work is interesting and promising, and the technology used has a chance for its further development, mainly in terms of reducing its costs as well as designing a dedicated device that allows for reliable reading of the results obtained (currently, a gel reader coupled with a smartphone is needed). Below are comments that, in my opinion, may have a positive impact on the final manuscript.

Abstract

“These tags are designed to exhibit error tolerance and can be easily processed in the field using paper tickets that emit fluorescence, enabling convenient reading and decoding via a mobile phone.”

This sentence should be rewritten as it suggests that a mobile phone is sufficient for reading and decoding. Meanwhile, the most important thing is gel reader which can measure fluorescence emitted.

Introduction

“The global food supply chain has over the past decade faced significant disruption. These have prompted efforts, from regulatory bodies like the UN”

The full name of the UN should be provided as it is mentioned for the first time in the manuscript.

I think that in the introduction the authors should also give some insights for currently existing commercial methods of DNA-based tagging and why they put an effort to establish a new one. Presently, the authors just mentioned UPC, QR, RFID, and NFC methods.

Figure 1.

(a)complexes. Need to be change to complexes.

Methods

“To minimize unintended interactions, we designed the barcode domains < Si > to be orthogonal using the following in silico process.”. “in silico” need to be write in italics.

“Random sequences at a length of 20 nucleotides (nt) were generated using a three-letter code (A, C, and T) to reduce secondary structures, with 30% to 70% GC-content to ensure similar melting temperatures”. The authors first mentioned that sequences are built from A, C, and T, and then provided GC-content, while the G is absent in the sequences. I think, this part need to be clarify.

Also in the Supp. material I found the table with sequences which were used in the experiments. This table should be labelled.

“The number of orthogonal sequences can be scaled up by using the enormous sequence diversity of 20-bp sequences (420)” Please put dot at the end of the sentence.

“Nitrocellulose (NC) paper was used as the substrate for the reporter ticket because it easily binds DNA molecules [?, 23, 33].” The reference should be checked once again since there is “?”.

“We baked the printed NC paper at 90 °C for 5 minutes, during which the wax melts and penetrates through the paper” This part needs to be checked again because here there is an information that baking time was 5 minutes, while in the supplementary material it is 10 minutes as follows “ The printed paper was then baked at 90°C for 10 minutes and cut into individual tickets. ”

“Each bit of DNA reporters consists of three oligonucleotide strands (Fig. 2, S1 (b)).” Please rewrite it as follows Fig. 2, and Fig. S1 (b), as it can be confusing for the readers.

Results

Fig 1. (b). I think that explanations for following abbreviations of T, U, and Si should me implemented. While Si is mentioned in the text above, there is a lack of information about T and U. Also, I believe that there should be one “=” sign instead of two.

Fig 1. (d). “(d) HyEn enables forgery protection.” Short explanation for both >TM, and sequencing scenarios should be provided to make it more clear for readers.

“Fig. 2 (a) shows the workflow of DNATrack system. Digital information (dataword) are encoded with error correction codes as a codeword.” The abbreviation ECC for error correction codes should be provided here since the authors use it after a

few times, and the abbreviation is much further in the "Encoding DNATags with error correction code" paragraph.

Fig. 1 C. "During the readout process after the ticket is exposed to the DNATag, the spots corresponding to digits 1s will turn darker, while the spots corresponding to 0s remain bright." Since the spots corresponding to digit 0s remain bright as all spots at the initial step, how the authors can prove that the ticket will not give you sometimes false positive results (in scenario that there will be no hybridization, spots remain bright, miming digit 0s there).

Figure 2. The size of the font should be checked in the whole figure since I got the impression it is not the same, for example (b).

Figure 2. c. Please explain what is X.

Figure 2. e. Abbr. for both Or and water should be included in the picture.

Figure 3. a. I believe there is a mistake in calculation. It should be 31 (45-14) instead of 39 in the upper left square, unless there is also a mistake in calculation of quantification of fluorescence. It needs to be carefully checked once again.

"In the first stage, two greyscaled fluorescence images from the reporter ticket – one pre-reaction and on post-reaction". Should be one post-reaction.

"I. Tag Amplification Attacks" and "II. Read-and-Rewrite Attacks". I believe these two sections are more fitting to the Discussion section rather than to the Results.

Table 1. The title should give more information about the content of the table. Its comparison of different tagging methods.

References

References need to be re edited according to the journal requirements since the consistency is not kept. Sometimes authors provide full names and surnames, and sometimes not. For example:

[1] Mayuri Joshi Shon Lesly Menezes Ramalingam H.M Aditya Umesh Shet, Sheikh Rameez Ibrahim. Implementation of efficient inception v2 model for apparel counterfeit detection. International Journal of Innovative Science and Research Technology, 7(4), April 2022.

[3] Kimberly L. Berk, Steven M. Blum, Vanessa L. Funk, Yuhua Sun, In-Young Yang, Mark V. Gostomski, Pierce A. Roth, Alvin T. Liem, Peter A. Emanuel, Michael E. Hogan, et al. Rapid visual authentication based on DNA strand displacement. ACS Applied Materials & Interfaces, 13(16):19476–19486, 2021.

Sometimes authors provide in italics the name of the journal, and sometimes the title of the paper. For example:

4] Madeleine S. Bloch, Daniela Paunescu, Philipp R. Stoessel, Carlos A. Mora, Wendelin J. Stark, and Robert N. Grass. Labeling milk along its production chain with DNA encapsulated in silica. Journal of agricultural and food chemistry, 62(43):10615–10620, 2014.

[6] Robert H Carlson. Biology is technology: the promise, peril, and new business of engineering life. Harvard University Press, 2011.

Furthermore, the authors give as references information that come from webpages which, are not a reliable scientific source, for example:

[7] Merlin Crossley. Is eating DNA safe? Online available at <https://theconversation.com/iseating-dna-safe-21016>, 2013.

[27] Laura Reiley. Meat processing plants are closing due to covid-19 outbreaks. beef shortfalls may follow. Online available at <https://www.washingtonpost.com/business/2020/04/16/meatprocessing-plants-are-closing-due-covid-19-outbreaks-beef-shortfalls-may-follow/>, 2020.

[31] https://en.wikipedia.org/wiki/Binary_Golay_code. Wikipedia: Binary golay code for error correction

[32] https://en.wikipedia.org/wiki/Linear_code. Wikipedia: Linear error correcting codes.

(Remarks on code availability)

Reviewer #3

(Remarks to the Author)

The manuscript by Li et al. describes a method DNA based product authentication method, based on a new detection scheme. The manuscript is an improvement of a method described by K. L. Berk (reference 3 of the manuscript), promising an improved security ("anti-forgery").

While the improved security features are interesting and novel, there is a lack of data surrounding the performance of this novelty. Rather the authors present application level experiments, which to a great extent so not seem technical feasible at a larger scale. Instead of discussing the novelty and performance details of the method (on a physio-chemical basis), the authors try to sell the applicability of the method in food tracing, which seems a very weak fit to the technology (see below). Also, the "anti-forgery" properties of the proposed method are significantly lower than described in the manuscript.

The level of performed experiments (e.g. replicability and reliability) is low, and the experimental description requires

improvement.

Details:

The method per se: The method (partially double stranded complexes) is inventive, however from the manuscript text and schemes the novelty is difficult to extract (and understand). Also, how the method differs from the state of the art (reference 3), and what the novelty is behind this step should be further emphasised.

a) Stability:

As the authors write, the method (partially double stranded complexes) has a low stability at elevated temperatures, so an important question is how stable the complexes are at room temperature. The only data presented are up to 7 days (with already a significant loss of signal), and consequently the applicability to longer time durations is questionable. (Figure S4c should also comprise quantitative values, not only color coding). More data is required on the stability of the tracers in more useful timescales and scenarios.

What is the life-time of the DNA bits in solution? How is this stability affected if other bits are also present in the mix (i.e. exchange of the double stranded section between the bits)? How is this affected if the DNA is stored dry? How is this affected if stored in the presence with trehalose?

b) Amounts of DNA required:

A very large disadvantage of the method is that it can not be combined with PCR for signal amplification, which is one of the main advantages of using DNA as a taggant. Consequently, very high amounts of DNA are required in the tagging. These limitations require discussion in the manuscript. Our calculations conclude that several μg of DNA are required per tag and spot. While there might be applications where this is feasible (e.g. microdots), the application shown (tracing of salad) does seem to be a very bad fit, as enormous amounts of DNA would be required to mark a whole salad head, or the location of the DNA on the salad would have to be known (which does not seem feasible in this application). The authors mention in the discussion that large amounts of DNATags are needed and a more quantitative discussion of this is required (i.e. how many μg of DNA required in a detection experiment). Also, it is unclear if per spot only one detection experiment can be performed? Could lower DNA amounts be traded off with longer detection times?

c) Forgeability:

The authors write, that the novel tracers can not be forged, however it is unclear why the following attack is not possible:

1. Extract DNA from product
2. split into two fractions
3. Sequence the first fraction to get all the sequences present in the taggant
4. Synthesise reporter oligos to all sequences identified, and create a "self-made" ticket, according to the description in the manuscript
5. Test the second fraction of DNA on the "self-made" ticket to identify which of the sequences present in the taggant are blocked to reveal the codeword

It is evident that this approach is much more elaborate than forging a standard DNA barcode, but it does not seem unforgeable, and also comes with significant disadvantages compared to a standard DNA barcode (not amplifiable).

In short, the method is not as non-forgeable as described, the stability of the taggants for over a week is unclear, and very high amounts of taggants are required per spot. While the idea is innovative, and an improvement over reference 3, the manuscript neither goes into the physio-chemical performance of the original idea, nor is the application level data shown convincing (and much thereof is hidden in the Supp. Info).

Also, the manuscript has a structural problem, as the last page of the introduction is rather a conclusion, which comprises some greatly overstated aspects of the technology, which are not supported by the data and or describe the work performed. (i.e. protection from amplification attacks, "significantly improved stability", "we....systematically assess reaction latency and error rates"). This section of the manuscript requires shortening, and the absence of any performance and conclusion statements (as these should follow the experiments and discussion). This section should rather be used to pose a hypothesis, which is then proven or rejected in the following sections of the manuscript.

More minor details:

The data quality of FIG 2f (far right) is very low. Is this from only one experiment, and why is the fluorescence decay shown on a log10 scale?

Fig 3c: The fit to the normalised data for the 0 bit is very poor. A better description is required (and might be found in bivariate normal distributions with Fisher transformation)

The experimental description is short, and the reporter complex (dye and quencher) are not described. Also the quenchers are not mentioned in the sequence listing. Neither is the application and removal from the cork and salad not described. More specification on "orange filter window" is required. Also, the experimental section to the device presented in Figure S11 is missing.

some typos: "This has" instead of "This have" (second sentence); "carring" instead of "carrying"; "Ref [?, 23, 33]", one

reference=?

(Remarks on code availability)

Reviewer #4

(Remarks to the Author)

(Remarks on code availability)

Version 1:

Reviewer comments:

Reviewer #1

(Remarks to the Author)

I am generally satisfied with the additional data provided and the text changes made in the manuscript. Since readers may have similar questions to mine, please add text to briefly describe the -20°C storage results (Figure S9) in the main manuscript, and also include some discussions of the previous questions 4 and 6 (natural DNA interference and readout turnaround time).

(Remarks on code availability)

Reviewer #2

(Remarks to the Author)

The authors improved manuscript based on my review report and all my concerns have been addressed. Therefore, I recommend this manuscript for publication.

(Remarks on code availability)

Reviewer #3

(Remarks to the Author)

The authors have significantly reworked the manuscript and have taken this reviewers comments into account.

While I still find the explanation of the scheme complicated to comprehend, this might be unavoidable (due to the nature of cryptographic schemes). I propose publishing the manuscript in the present form.

(Remarks on code availability)

Reviewer #4

(Remarks to the Author)

(Remarks on code availability)

Author's Response to Reviews of:

Hybridization-Encoded DNA Tags with Paper-Based Readout for Anti-Forgery Raw Material Tracking

Response to Reviewer #1:

The manuscript by Li et al. presented a very interesting DNA hybridization-based method for encoding and tracking in the food supply chain. Compared to the standard labels on the outer packaging, DNA oligonucleotides can be directly applied to the surface of the raw material, thus are more difficult to remove or change. In the DNA oligonucleotide "tags", each digit was represented by DNA hybridization status (single-stranded = 1, double-stranded = 0) instead of by base sequences. Therefore, the encoded information can be easily erased by heating, and difficult to replicate or decode by PCR or sequencing. The authors developed an integrated system for tracking, and tested the feasibility on different types of surfaces. The cost for labeling and readout was reasonable, and the process was not too complex. This is an inspiring application of DNA-based information storage and DNA hybridization.

We thank the reviewer for their careful reading and accurate summary of our work.

The manuscript is well written, and the figures are clearly presented. I have a few questions for the authors, and additional experimental data might be needed to enhance the depth and clarity of this work:

1. In non-cold chain transportation, the tagged material might experience medium-high temperature in tropical/subtropical areas, and even in temperate zone during summer. Will the encoded information be erased at 30-60 °C during shipping? Can this method be applied in cold chain transportation, especially in frozen status? Please discuss the temperature tolerance of the hybridization-based DNA tags.

We thank the reviewer for their careful reading and thoughtful advice. To respond to this comment, currently the DNA bit sequences are designed with a melting temperature of 60°C, ensuring their stability for room-temperature uses. For applications requiring higher temperatures, the universal domain U can be extended to increase the melting temperature. We added the following statements in the 'Molecular design and reaction assessment' section:

"The T domain serves as a toehold for initiating strand displacement reaction during the readout, while the U* domain provides stability of the partial double-strand structure at room temperature (Fig. S9). For applications requiring higher ambient temperatures, the universal domain U can be elongated to increase the melting temperature (T_m) (Fig. S16)."*

In terms of frozen status, we added experimental results testing the stability of DNATags stored at -20°C in Figure S9. The results showed that our DNATags stayed stable after being frozen. We are attaching a snapshot here for convenience:

(d) Concentration of the DNA Bit 1 or denatured Bit 0

2. The DNA tags might be applied on dry or wet surfaces. In dry conditions, how long can the double-stranded DNA last without losing its secondary structure? In wet conditions, since the single-stranded and double-stranded DNA probes seem to have the same sequence in "T" and "U" domains in Figure 2, the "TU" oligos may slowly shift to the single strands after long storage or transportation, making it difficult to distinguish 0 and 1 signals. How long can the tags work properly?

Response: This is a great question, and we appreciate the reviewer's thoughtful consideration. Due to the 60°C melting temperature of our DNA complex, it is incompatible with accelerated aging studies, which typically operate at temperatures above 65°C .

Therefore, we cannot directly measure the long-term stability of our dsDNA complex. To address this, we referenced findings from previous studies on DNA stability for long-term storage. Additionally, we conducted an ODE simulation to assess the potential kinetic leakage of DNA bits "1" and "0" coexisting in solution, as shown in Figure S15. The results demonstrated that our DNATags remain stable for over 10 years.

These findings have been included in the *'Reliability and End-to-End Testing of DNATrack'* section, with the relevant text shown below. For convenience, we also provide partial snapshots of Figure S15:

"To better understand the long-term stability of DNATags, we focus on two primary factors: DNA strand damage and bit-exchange leakage. DNA strand stability has been extensively studied in DNA storage research [24,25]. Organick et al. reported that DNA molecules are least stable in solution, with a half-life of approximately one year. However, this half-life can be extended to over 10 years when stored in trehalose and may exceed 100 years under 'dry' conditions, depending on the storage methods used [25].

Beyond strand damage, bit-exchange leakage is another key consideration for DNATag stability. Since our DNATags encode information in metastable states (i.e., partially double-stranded and single-stranded DNA), bit-1s and bit-0s gradually rehybridize to reach their equilibrium state over time. The pathways for bit-exchange are illustrated in Fig. S15 (a). Leveraging the extensively studied thermodynamics of DNA hybridization in solution [16,34], we simulated these exchange pathways and found that DNATags remain stable with minimal kinetic leakage in solution for over 10 years (Fig. S15 (b)). Under dry conditions, we tested the stability of dried DNATags for up to 15 days and all DNATags stayed stable after the storage (Fig. S8). For longer term, we expect the bit-exchange leakage to be lower than in solution due to the significantly reduced mobility of DNA molecules at dry state.

(b) ODE Simulation of Kinetic Leakage of DNA Bits in Solution

3. Will the DNA tags be sabotaged or contaminated by rubbing or washing?

Response: We thank the reviewer for bringing up this question. Currently we have not built up any protection of DNATags against physical sabotage, rubbing, or washing. DNATags should not be easily damaged by physical force from rubbing but do suffer from being removed by either rubbing or washing. Protection of DNATag against it is a future direction that we are working on, and we have added relevant discussions in the Discussion section, quoted here for convenience:

“Future developments aim to expand DNATags’ compatibility with various objects. Current DNATags are designed for attachment to solid surfaces but may be easily washed away. Research into waterproof coatings could prevent damage from washing or abrasion while maintaining rehydration ease for readout. Moreover, attaching DNATags to silica or magnetic nanoparticles and encapsulating them in silica could extend their utility, enabling applications in labeling fabrics, liquids, and other diverse objects [12, 17, 26, 31].”

4. Since plant or animal-derived raw material have intrinsic nucleic acids, will these DNA or RNA affect the readout?

Response: Natural DNA is unlikely to interfere with our DNA reporter detection due to the high selectivity of strand displacement-based reporters, where even a few base mismatches drastically reduce hybridization efficiency. With a 35-nt displaced domain, the probability of a natural DNA/RNA matching our DNABit is extremely low. Furthermore, most natural DNA exists in double-stranded form, which does not interact with our reporters that target single-stranded DNA. Additionally, RNA is typically unstable outside of cells. As a result, natural nucleic acids are not expected to impact the readout.

5. The limitations of this tagging method need to be discussed in detail, including types of compatible material, required shipping and storage conditions, etc.

Response: We thank the reviewer for pointing this out, and we have added a more thorough discussion of limitations including the requirement of large DNA quantity, and compatible materials.

“Despite its advantages, DNATrack has limitations. One major challenge is the significant DNA quantity required for a single readout (~60 µg per destructive readout), primarily due to the large reaction buffer (60 µL) needed to cover the DNATag-reading ticket (~1 cm²). Since reaction speed is semi-proportional to DNA concentration, the large buffer necessitates higher DNA inputs to ensure efficient detection. To address this, smaller DNATag-reading tickets with reduced size of reporter spots can be produced using a microarray printer, or the reaction time can be extended, as signal intensity is semi-proportional to both DNA concentration and reaction time.

Future developments aim to expand DNATags’ compatibility with various objects. Current DNATags are designed for attachment to solid surfaces but may be easily washed away. Research into waterproof coatings could prevent damage from washing or abrasion while maintaining rehydration ease for readout. Moreover, attaching DNATags to silica or magnetic

nanoparticles and encapsulating them in silica could extend their utility, enabling applications in labeling fabrics, liquids, and other diverse objects [12, 17, 26, 31]. Lastly, while DNATags improve security, further investigation is needed to address potential forgery attacks. With the increasing accessibility of DNA amplification and sequencing, identifying and mitigating vulnerabilities is crucial to ensure DNATrack's long-term effectiveness."

For storage conditions, we have added the following paragraphs in the section of "Reliability and end-to-end testing of DNATrack."

"To better understand the long-term stability of DNATags, we focus on two primary factors: DNA strand damage and bit-exchange leakage. DNA strand stability has been extensively studied in DNA storage research [25, 26]. Organick et al. reported that DNA molecules are least stable in solution, with a half-life of approximately one year. However, this half-life can be extended to over 10 years when stored in trehalose and may exceed 100 years under 'dry' conditions, depending on the storage methods used [26].

Beyond strand damage, bit-exchange leakage is another key consideration for DNATag stability. Since our DNATags encode information in metastable states (i.e., partially double-stranded and single-stranded DNA), bit-1s and bit-0s gradually rehybridize to reach their equilibrium state over time. The pathways for bit-exchange are illustrated in Fig. S15 (a). Leveraging the extensively studied thermodynamics of DNA hybridization in solution [16, 37], we simulated these exchange pathways and found that DNATags remain stable with minimal kinetic leakage in solution for over 10 years (Fig. S15 (b)). Under dry conditions, we tested the stability of dried DNATags for up to 15 days and all DNATags stayed stable after the storage (Fig. S8). For longer term, we expect the bit-exchange leakage to be lower than in solution due to the significantly reduced mobility of DNA molecules at dry state."

6. This method currently requires 10-30 min turnaround time. Is it possible to speed it up by optimizing the readout reactions and processes?

Response: Yes. We did not focus on pushing the limit of minimal turnaround time because our priority was to first build a complete, easy-to-use, end-to-end system. According to the kinetics test result (Figure 1 (f)), a reaction time of 2 minutes already yielded a significant signal, so a turnaround time of 2 minutes is feasible. Presumably, the turnaround can be further shortened below 2 minutes by various strategies, e.g., utilizing different sets T domain sequences for less kinetic interruption between different DNA bits, and selecting a better fluorophore that yields a higher signal-to-noise ratio.

Minor points:

1. Description of the error correction code is a bit hard to follow. Additional description is needed.

Response: We thank the reviewer for pointing it out. We have added a more detailed description of the error correction code:

"Writing and reading DNATags can introduce errors due to factors like DNA synthesis and inherently imprecise of pipetting. To ensure accurate and robust decoding, we employ error

correction code (ECC), which are techniques for detecting and correcting errors in data. Our approach uses a linear code, which minimizes redundant bits while still allowing error correction in short messages [11,13,31].

In this process, we start with the binary dataword (the original information we want to encode) and multiply it by a generator matrix. This transformation yields a longer codeword, which includes both the original data and additional redundant bits specifically designed to detect and correct errors. These redundant bits provide tolerance for up to a few errors in the codeword. The codeword is then directly encoded by DNATag and when it is read by the ticket, our ECC-integrated decoder uses these redundant bits to retrieve the original dataword, correcting any errors that may have occurred (Fig. 2(a)). The amount of redundancy (number of redundant bits) can be adjusted based on the application's error tolerance and expected error rate. In our specific demonstration, we chose a 12-bit dataword and a 24-bit codeword (Fig. 3(d)), giving us the ability to correct errors of up to 3 bits in the codeword readout."

2. *There are a few typos in the manuscript, such as "This work proposes the use DNA, with..." in the Abstract.*

Response: We thank the reviewer for help pointing this out. We have carefully checked and corrected the typos in the manuscript.

3. *Please check the reference numbers, such as "Nitrocellulose (NC) paper was used as the substrate for the reporter ticket because it easily binds DNA molecules [?, 23, 33]." in the Methods section.*

Response: We thank the reviewer for help pointing this out. We have corrected the reference numbers.

Response to Reviewer #2:

The publication describes a new system for labeling physical objects called DNATrack employs specially designed DNA mixtures, termed DNATags (made of single-stranded and partially double-stranded DNA), for product tracking. The authors described in detail the experiments performed, as well as compared the cost of the proposed technology with others described in the scientific literature. I think that the work is interesting and promising, and the technology used has a chance for its further development, mainly in terms of reducing its costs as well as designing a dedicated device that allows for reliable reading of the results obtained (currently, a gel reader coupled with a smartphone is needed).

We thank the reviewer for their careful readthrough and helpful comments.

Below are comments that, in my opinion, may have a positive impact on the final manuscript.

Abstract

"These tags are designed to exhibit error tolerance and can be easily processed in the field using paper tickets that emit fluorescence, enabling convenient reading and decoding via a mobile phone. "

This sentence should be rewritten as it suggests that a mobile phone is sufficient for reading and decoding. Meanwhile, the most important thing is gel reader which can measure fluorescence emitted.

Response: We thank the reviewer for the helpful suggestions. We have re-written this sentence, so it conveys the information more clearly:

“These tags are designed to exhibit error tolerance and can be easily processed in the field using paper tickets that emit fluorescence, enabling convenient reading and decoding via a mobile phone and a field-deployable fluorescence filtering device.”

Introduction

“The global food supply chain has over the past decade faced significant disruption. These have prompted efforts, from regulatory bodies like the UN”

The full name of the UN should be provided as it is mentioned for the first time in the manuscript.

We thank the reviewer for pointing these out. We have replaced “UN” with its full name “United Nations”.

I think that in the introduction the authors should also give some insights for currently existing commercial methods of DNA-based tagging and why they put an effort to establish a new one. Presently, the authors just mentioned UPC, QR, RFID, and NFC methods.

Response: We have also added details in the Introduction section about commercial methods of DNA-based tagging schemes and the technology gap.

“In the realm of supply chain tracking, the ideal design for DNA tags involves scalability, affordability, field adaptability, robustness, and anti-forgery features. Recent studies have made strides in enhancing DNA stability using methods like silica nanoparticle or microbial spore encapsulation [15, 22, 32, 39]. Cost-effective reading and writing methods, including nanopore sequencers and paper ticket systems, have also been explored [2, 9]. Beyond scientific research, several commercial enterprises have developed DNA-based tagging technologies [19]. For instance, Haelixa employs silica-encapsulated DNA tags that can adhere to tangible goods [39]. CypherMark’s TraceTag technology uses a pair of primers as detection keys, enabling quantitative PCR-based identification for applications like cash security and anti-counterfeiting [30, 36]. Similarly, Tagsmart’s Smart DNA Tags integrate synthetic DNA tags with an online platform for artwork certification [18], while Anika Biosciences utilizes bacterial spore-protected microbial tags for food labeling [38]. Despite these innovations, most existing solutions tend to address isolated aspects of DNA tagging rather than offering a comprehensive system design for large-scale functionality. Notably, the widespread replication of DNA has been overlooked, and our research exposes vulnerabilities in existing schemes to forgery through simple laboratory protocols, emphasizing the need for a holistic approach in designing effective tagging systems.”

Figure 1.

(a)complexs. Need to be change to complexes.

Response: We thank the reviewer for pointing this out. We have corrected the typo.

Methods

“To minimize unintended interactions, we designed the barcode domains < Si > to be orthogonal using the following in silico process.”. “in silico” need to be write in italics.

Response: We thank the reviewer for pointing this out. We have corrected the font.

“Random sequences at a length of 20 nucleotides (nt) were generated using a three-letter code (A, C, and T) to reduce secondary structures, with 30% to 70% GC-content to ensure similar melting temperatures”. The authors first mentioned that sequences are built from A, C, and T, and then provided GC-content, while the G is absent in the sequences. I think, this part need to be clarify.

Response: We thank the reviewer for the comment. We have added clarification on that in the main text:

“Random sequences at a length of 20 nucleotides (nt) were generated using a three-letter code (A, C, and T) to reduce secondary structures, with 30% to 70% GC-content (note: now there is only 'C' in the sequence that makes up the 'GC-content') to ensure similar melting temperatures;”

Also in the Supp. material I found the table with sequences which were used in the experiments. This table should be labelled.

Response: We have labeled the table as ‘Table S1’ in our main text.

“The number of orthogonal sequences can be scaled up by using the enormous sequence diversity of 20-bp sequences (4^{20})” Please put dot at the end of the sentence.

Response: We thank the reviewer for pointing this out and we have corrected the typo.

“Nitrocellulose (NC) paper was used as the substrate for the reporter ticket because it easily binds DNA molecules [?, 23, 33].”The reference should be checked once again since there is “?”.

Response: We thank the reviewer for pointing this out and we have corrected the reference number.

“We baked the printed NC paper at 90 °C for 5 minutes, during which the wax melts and penetrates through the paper” This part needs to be checked again because here there is an information that baking time was 5 minutes, while in the supplementary material it is 10 minutes as follows “ The printed paper was then baked at 90°C for 10 minutes and cut into individual tickets. ”

Response: We thank the reviewer for pointing out the inconsistency in our writing. In experiments we fabricated several batches of the NC paper tickets. We initially baked them for 5 minutes and it worked well, but later we decided to change it to 10 minutes for presumably better consistency because we found it took the oven more than 1 minute to get back to 90°C after putting the tickets inside. Empirically we found no difference between the two baking processes (either by visual inspection of the ticket or by inspecting the readout

signal). The materials of SI and the main text were written at different timepoints and thus the difference. We have updated the materials and used 10 minutes to clear the confusion.

“Each bit of DNA reporters consists of three oligonucleotide strands (Fig. 2, S1 (b)).” Please rewrite it as follows Fig. 2, and Fig. S1 (b), as it can be confusing for the readers.

Response: We thank the reviewer for the suggestion. We have added the following paragraph and sentences following Fig. 2 to make it easier to follow and understand:

“In this method, a DNA Bit 1 is single-stranded DNA strand, $\langle T^*U^*Si^* \rangle$, and a DNA Bit 0 is a partially double-stranded DNA complex, where the strand $\langle T^*U^*Si^* \rangle$ is bound to a blocker strand $\langle UT \rangle$ (Fig. 1(b)). The T^* domain serves as a toehold for initiating strand displacement reaction during the readout, while the U^* domain provides stability of the partial double-strand structure at room temperature (Fig. S9). For applications requiring higher ambient temperatures, the universal domain U can be elongated to increase the melting temperature (T_m) (Fig. S16). All DNA Bits share the same sequence in the $\langle T^*U^* \rangle$ domains, but each Bit carries a unique Si^* domain ($i = 1, 2, 3, \dots$), corresponding to a distinct digit in the codeword. This hybridization encoding method effectively mitigates forgery risks. During a PCR attack, the high-temperature denaturation step dissociates the DNA duplexes representing DNA Bit ‘0’s, rendering the encoding irretrievable and preventing recovery of the original information (Fig. 1(d)). Similarly if an attacker sequences the DNATag, they can determine only the individual sequences used in the DNA Bits, but not the hybridization states encoding the binary digits.

Reading DNATags utilizes toehold mediated strand displacement on a nitrocellulose paper ticket, producing a fluorescent pattern [2]. A set of fluorophore-labeled DNA reporter complexes containing different Si domains are arrayed on a nitrocellulose paper ticket. Each reporter complex consists of a probe strand $\langle XSjUT \rangle$, an anchor strand $\langle U^*Sj^* \rangle$, and a fluorophore strand $\langle FX^* \rangle$ (Fig. 2 (C)). The anchor strand contains a 30-nt poly-T tail on its 3’ end for enhanced binding to the nitrocellulose paper (Fig. S3). Upon contact, DNA Bit 1 strand $\langle T^*U^*Si^* \rangle$ can bind the toehold domain of the probe strand and subsequently displace the probe strand together with the fluorophore strand off the anchor strand if the Si^* matches the Sj domain (i.e. $i = j$). The displaced fluorophore strands can be easily washed away, resulting in significantly reduced fluorescent signals. In contrast, the ‘0’s of DNA Bits cannot displace the probe strand because the toehold region $\langle T \rangle$ is covered by a blocker strand, resulting in unchanged high fluorescence. Thus, the DNATag reading ticket initially displays an array of uniformly bright fluorescent dots, and it switches to a specific pattern of bright and dim spots upon exposure to a DNATag.”

We also edited the following paragraph in Method section, so it aligns better with the previous paragraphs:

“Each bit of DNA reporters consists of three oligonucleotide strands including $\langle FX^* \rangle$, $\langle 30TU^*Si^* \rangle$, and $\langle X Si U T \rangle$ (Fig. 2 (c), S1 (b)). To assemble the reporter complex, we mixed the oligonucleotide strands at a final concentration of 2 μ M of $\langle FX^* \rangle$, 2.2 μ M of $\langle 30T U^*Si^* \rangle$, and 2.4 μ M of $\langle X Si U T \rangle$ in 1x Tris-EDTA (TE) buffer with 10 mM $MgCl_2$.”

Results

Fig 1. (b). I think that explanations for following abbreviations of T, U, and Si should be implemented. While Si is mentioned in the text above, there is a lack of information about T and U. Also, I believe that there should be one "=" sign instead of two.

Response: We thank the reviewer for the suggestion, and we have added more description of the different domains and their functions, as quoted in the previous response. We have also corrected the '=' sign in Figure 1.

Fig 1. (d). "(d) HyEn enables forgery protection." Short explanation for both $>^{\text{TM}}$, and sequencing scenarios should be provided to make it more clear for readers.

Response: We thank the reviewer for the suggestion, and we have added explanations to it in the first paragraph of 'Molecular design and reaction assessment' section:

"This hybridization encoding method effectively mitigates forgery risks. During a PCR attack, the high-temperature denaturation step dissociates the DNA duplexes representing DNA Bit '0's, rendering the encoding irretrievable and preventing recovery of the original information (Fig. 1 (d)). Similarly if an attacker sequences the DNATag, they can determine only the individual sequences used in the DNA Bits, but not the hybridization states encoding the binary digits."

"Fig. 2 (a) shows the workflow of DNATrack system. Digital information (dataword) are encoded with error correction codes as a codeword." The abbreviation ECC for error correction codes should be provided here since the authors use it after a few times, and the abbreviation is much further in the "Encoding DNATags with error correction code" paragraph.

Response: We thank the reviewer for pointing it out and we have replaced it with the abbreviation 'ECC'.

Fig. 1 C. "During the readout process after the ticket is exposed to the DNATag, the spots corresponding to digits 1s will turn darker, while the spots corresponding to 0s remain bright." Since the spots corresponding to digit 0s remain bright as all spots at the initial step, how the authors can prove that the ticket will not give you sometimes false positive results (in scenario that there will be no hybridization, spots remain bright, miming digit 0s there).

Response: Indeed. One easy strategy to screen out the mentioned 'false positive' results is to add one set of reporters (i.e., one spot on the reading ticket) as a positive control. The corresponding DNA Bit 1 strand would be in all DNATags despite the dataword. Thus, if the positive control spot went dim, that indicates a successful readout of the DNATag. Otherwise, it would be a false result.

Figure 2. The size of the font should be checked in the whole figure since I got the impression it is not the same, for example (b).

Response: We thank the reviewer for pointing this out. We have updated all figures, so the fonts are of the same size.

Figure 2. c. Please explain what is X.

Response: We thank the reviewer for pointing this out. We have added a thorough explanation of all domains in the beginning of 'Molecular design and reaction assessment' section, as quoted in earlier response.

Figure 2. e. Abbr. for both Or and water should be included in the picture.

Response: We thank the reviewer for the suggestion. We have included abbreviation of both Or and water in the figure.

Figure 3. a. I believe there is a mistake in calculation. It should be 31 (45-14) instead of 39 in the upper left square, unless there is also a mistake in calculation of quantification of fluorescence. It needs to be carefully checked once again.

Response: We thank the reviewer for pointing out the miscalculation. We have checked and corrected the calculations in the demonstration diagram.

"In the first stage, two greyscaled fluorescence images from the reporter ticket – one pre-reaction and on post-reaction". Should be one post-reaction.

Response: We thank the reviewer for pointing this out. We have corrected the typo.

"I. Tag Amplification Attacks" and "II. Read-and-Rewrite Attacks". I believe these two sections are more fitting to the Discussion section rather than to the Results.

Response: We thank the reviewer for the suggestion. After attempts at reorganizing and relocating these sections into Discussion section, we still find it more appropriate to place them at the end of the 'Result' section. The reason is because the 'Tag Amplification Attacks' subsection mainly describes the results shown in Figure 5 and Figure S12, and the 'Read-and-Rewrite Attacks' subsection serves as an additional but equally important explanation of the result. We consider these results to be of importance since it showcases the unique security feature of DNATag, so we would keep these subsections in 'Result' section.

Table 1. The title should give more information about the content of the table. Its comparison of different tagging methods.

Response: We thank the reviewer for the suggestion. We have changed the table name to: "Comparison between existing implementations of DNA tagging schemes".

References

References need to be re edited according to the journal requirements since the consistency is not kept. Sometimes authors provide full names and surnames, and sometimes not. For example:

[1] Mayuri Joshi Shon Lesly Menezes Ramalingam H.M Aditya Umesh Shet, Sheikh Rameez Ibrahim. Implementation of efficient inception v2 model for apparel counterfeit detection. *International Journal of Innovative Science and Research Technology*, 7(4), April 2022.

[3] Kimberly L. Berk, Steven M. Blum, Vanessa L. Funk, Yuhua Sun, In-Young Yang, Mark V. Gostomski, Pierce A. Roth, Alvin T. Liem, Peter A. Emanuel, Michael E. Hogan, et al. Rapid visual authentication based on DNA strand displacement. *ACS Applied Materials & Interfaces*, 13(16):19476–19486, 2021.

Sometimes authors provide in italics the name of the journal, and sometimes the title of the paper. For example:

4] Madeleine S. Bloch, Daniela Paunescu, Philipp R. Stoessel, Carlos A. Mora, Wendelin J. Stark, and Robert N. Grass. *Labeling milk along its production chain with DNA encapsulated in silica*. *Journal of agricultural and food chemistry*, 62(43):10615–10620, 2014.

[6] Robert H Carlson. *Biology is technology: the promise, peril, and new business of engineering life*. Harvard University Press, 2011.

Furthermore, the authors give as references information that come from webpages which, are not a reliable scientific source, for example:

[7] Merlin Crossley. *Is eating DNA safe?* Online available at <https://theconversation.com/iseating-dna-safe-21016>, 2013.

27] Laura Reiley. *Meat processing plants are closing due to covid-19 outbreaks. beef shortfalls may follow*. Online available at <https://www.washingtonpost.com/business/2020/04/16/meatprocessing-plants-are-closing-due-covid-19-outbreaks-beef-shortfalls-may-follow/>, 2020.

[31] https://en.wikipedia.org/wiki/Binary_Golay_code. Wikipedia: Binary golay code for error correction

[32] https://en.wikipedia.org/wiki/Linear_code. Wikipedia: Linear error correcting codes.

Response: We thank the reviewer for careful readthrough and the suggestions. We have corrected the formats of the references and replaced or appended the webpage references with more peer reviewed scientific articles. We would also like to note that, the previous reference [6], now reference [5], ‘Biology is technology: the promise, peril, and new business of engineering’ is a book, so we are providing the book name in italics.

Response to Reviewer #3:

The manuscript by Li et al. describes a method DNA based product authentication method, based on a new detection scheme. The manuscript is an improvement of a method described by K. L. Berk (reference 3 of the manuscript), promising an improved security (“anti-forgery”).

While the improved security features are interesting and novel, there is a lack of data surrounding the performance of this novelty. Rather the authors present application level experiments, which to a great extent so not seem technical feasible at a larger scale. Instead of discussing the novelty and performance details of the method (on a physio-chemical basis), the authors try to sell the applicability of the method in food tracing, which seems a very weak fit to the technology (see below). Also, the “anti-forgery” properties of the proposed method are significantly lower than described in the manuscript.

Response: We thank the reviewer for their careful readthrough and enlightening comments. We have adjusted our writing so that the results are now presented in a better way and that our unique novelties are summarized more clearly. We have also added new experimental results validating the reliability and stability of DNATrack. These changes will be discussed in detail in the following responses.

-

R3.1: The level of performed experiments (e.g., replicability and reliability) is low, and the experimental description requires improvement.

Response: We appreciate the reviewer's comments on the replicability and reliability of our experiments. Below, we first discuss the sources of signal variability, and the measures implemented to improve consistency in our system. Additionally, we have revised and expanded the methods section to provide greater detail on our experimental procedures, including the specificity testing of DNA Bits, the process of reading DNATag solutions using the reading tickets, the process of labeling objects with DNATags and retrieving the DNATags for readout, and how we did the Gaussian fitting of our statistical data.

The DNATag platform relies on DNA strand displacement reactions paired with paper-based readout. While DNA strand displacement reactions are well-studied and recognized as highly predictable and deterministic, we acknowledge that the paper-based readout component introduces greater variability in signal measurements.

To assess and mitigate this variability, we conducted extensive testing, evaluating hundreds of reactions between DNA Bits and reporters (Fig. 3c). To improve reproducibility, we developed an in-silico signal processing pipeline that converts raw analog fluorescence signals into binary digital outputs. Furthermore, we implemented error correction codes (ECC) to address signal inconsistencies. This approach is analogous to digital systems, where noisy analog signals are processed with ECC to ensure robust and reliable performance.

For end-to-end validation, we observed that adding trehalose during storage significantly reduced errors. We performed many experiments under each condition, supported by statistical analysis to confirm the reliability of these findings (Fig. 4d).

In summary, while the paper-based readout exhibits inherent signal variability, our system compensates for this through comprehensive testing, statistical analysis, and the application of advanced signal processing techniques with ECC. These measures collectively enhance the overall reliability and reproducibility of the DNATag system.

To enhance the data quality, we performed new experiments with triplicates to for the reaction kinetics (Figure 2 (f)). We are attaching a snapshot of updated Figure 2 here for convenience of reading:

To further support the reliability of the system, we added an additional section for studying the long-term stability of DNATags. Additionally, we have re-structured our experimental

description (the 'Methods' section) for easier reading. The relevant text of the section “Reliability and end-to-end testing of DNATrack” is shown below for convenience.

“To better understand the long-term stability of DNATags, we focus on two primary factors: DNA strand damage and bit-exchange leakage. DNA strand stability has been extensively studied in DNA storage research [24,25]. Organick et al. reported that DNA molecules are least stable in solution, with a half-life of approximately one year. However, this half-life can be extended to over 10 years when stored in trehalose and may exceed 100 years under ‘dry’ conditions, depending on the storage methods used [25].

Beyond strand damage, bit-exchange leakage is another key consideration for DNATag stability. Since our DNATags encode information in metastable states (i.e., partially double-stranded and single-stranded DNA), bit-1s and bit-0s gradually rehybridize to reach their equilibrium state over time. The pathways for bit-exchange are illustrated in Fig. S15 (a). Leveraging the extensively studied thermodynamics of DNA hybridization in solution [16,34], we simulated these exchange pathways and found that DNATags remain stable with minimal kinetic leakage in solution for over 10 years (Fig. S15 (b)). Under dry conditions, we tested the stability of dried DNATags for up to 15 days and all DNATags stayed stable after the storage (Fig. S8). For longer term, we expect the bit-exchange leakage to be lower than in solution due to the significantly reduced mobility of DNA molecules at dry state.”

Snapshot of Figure S12, ODE simulation results.

Details:

The method per se: The method (partially double stranded complexes) is inventive, however from the manuscript text and schemes the novelty is difficult to extract (and understand). Also,

how the method differs from the state of the art (reference 3), and what the novelty is behind this step should be further emphasised.

Response: We thank the reviewer for the comment. We have edited the article to put emphasis on our novelties. Now we have the following paragraphs:

“Hybridization Encoding and reaction assessment. A common method for encoding information in DNA involves representing individual digital bits as the presence or absence of pre-specified DNA strands. These “DNA bits” are combined to encode arbitrary binary numbers [2, 9]. Berk et al. demonstrated that the presence of a pre-specified DNA strand could generate a fluorescent signal on a paper readout via toehold-mediated strand displacement reactions [2]. While this approach provides a cost-effective way to read DNA tags using paper-based readouts, it suffers from significant vulnerabilities to forgery. For instance, a hacker could amplify a DNA taggant via PCR or sequence the taggant to reproduce it.

To address this security vulnerability, we propose a hybridization-based encoding scheme for representing digital bits. In this method, a DNA Bit 1 is single-stranded DNA strand, $\langle T^*U^*Si^* \rangle$, and a DNA Bit 0 is a partially double-stranded DNA complex, where the strand $\langle T^*U^*Si^* \rangle$ is bound to a blocker strand $\langle UT \rangle$ (Fig. 1 (b)). The T^* domain serves as a toehold for initiating strand displacement reaction during the readout, while the U^* domain provides stability of the partial double-strand structure at room temperature (Fig. S9). For applications requiring higher ambient temperatures, the universal domain U can be elongated to increase the melting temperature (T_m) (Fig. S16). All DNA Bits share the same sequence in the $\langle T^*U^* \rangle$ domains, but each Bit carries a unique Si^* domain ($i = 1, 2, 3, \dots$), corresponding to a distinct digit in the codeword. This hybridization encoding method effectively mitigates forgery risks. During a PCR attack, the high-temperature denaturation step dissociates the DNA duplexes representing DNA Bit ‘0’s, rendering the encoding irretrievable and preventing recovery of the original information (Fig.1 (d)). Similarly if an attacker sequences the DNATag, they can determine only the individual sequences used in the DNA Bits, but not the hybridization states encoding the binary digits.”

a) Stability:

As the authors write, the method (partially double stranded complexes) has a low stability at elevated temperatures, so an important question is how stable the complexes are at room temperature. The only data presented are up to 7 days (with already a significant loss of signal), and consequently the applicability to longer time durations is questionable. (Figure S4c should also comprise quantitative values, not only color coding). More data is required on the stability of the tracers in more useful timescales and scenarios.

What is the life-time of the DNA bits in solution? How is this stability affected if other bits are also present in the mix (i.e. exchange of the double stranded section between the bits)? How is the affected if the DNA is stored dry? How is this affected if stored in the presence with trehalose?

Response: We thank the reviewer for highlighting these questions regarding the stability of DNATags. Due to the 60°C melting temperature of our DNA complex, it is incompatible with accelerated aging studies, which typically operate at temperatures above 65°C. Therefore, we

cannot directly measure the long-term stability of our dsDNA complex. To address this, we referenced findings from previous studies on DNA stability for long-term storage. Additionally, we conducted Ordinary Differential Equation (ODE) simulation and additional experiments to investigate DNATag stability. The following paragraph has been added to the article:

“To better understand the long-term stability of DNATags, we focus on two primary factors: DNA strand damage and bit-exchange leakage. DNA strand stability has been extensively studied in DNA storage research [24,25]. Organick et al. reported that DNA molecules are least stable in solution, with a half-life of approximately one year. However, this half-life can be extended to over 10 years when stored in trehalose and may exceed 100 years under ‘dry’ conditions, depending on the storage methods used [25].

Beyond strand damage, bit-exchange leakage is another key consideration for DNATag stability. Since our DNATags encode information in metastable states (i.e., partially double-stranded and single-stranded DNA), bit-1s and bit-0s gradually rehybridize to reach their equilibrium state over time. The pathways for bit-exchange are illustrated in Fig. S15 (a). Leveraging the extensively studied thermodynamics of DNA hybridization in solution [16,34], we simulated these exchange pathways and found that DNATags remain stable with minimal kinetic leakage in solution for over 10 years (Fig. S15 (b)). Under dry conditions, we tested the stability of dried DNATags for up to 15 days and all DNATags stayed stable after the storage (Fig. S8). For longer term, we expect the bit-exchange leakage to be lower than in solution due to the significantly reduced mobility of DNA molecules at dry state.”

It was surprising to observe noticeable signal loss in our 7-day experiment, given that DNA is known to have a half-life of several years. We believe this observed "signal loss" was likely due to fluctuations caused by handling variations. In that experiment, the reading tickets were manually prepared and subjected to two washing and drying steps, which could have resulted in the loss of DNA reporters during the washing process.

To determine whether DNATags genuinely degrade over such a short timescale, we conducted a follow-up experiment. Dry DNATags were stored for 15 days and quantified immediately after rehydration using a microplate reader. Unlike the previous method, this approach does not involve washing steps and offers much greater sensitivity. Our results showed that DNATags did not exhibit any signal loss over the 15-day period. We have included the results in Figure S8 and we attach a partial snapshot of the result here for convenience:

Caption: Concentrations of the DNA Bit 1 or denatured DNA Bit 0 calculated by linear fitting their induced fluorescence intensities to the positive and negative controls. The 0-day datapoints of Bit 1 were higher because the 0-day DNATags were directly added to the reporter plate after preparation, so they did not go through the drying and retrieval steps and thus did not have any loss from retrieval. According to the measurement, DNATags did not show any sign of degradation or denaturation at dry storage up to 15 days.

b) Amounts of DNA required:

A very large disadvantage of the method is that it can not be combined with PCR for signal amplification, which is one of the main advantages of using DNA as a taggant. Consequently, very high amounts of DNA are required in the tagging. These limitations require discussion in the manuscript. Our calculations conclude that several μg of DNA are required per tag and spot. While there might be applications where this is feasible (e.g. microdots), the application shown (tracing of salad) does seem to be a very bad fit, as enormous amounts of DNA would be required to mark a whole salad head, or the location of the DNA on the salad would have to be known (which does not seem feasible in this application). The authors mention in the discussion that large amounts of DNATags are needed and a more quantitative discussion of this is required (i.e. how many μg of DNA required in a detection experiment). Also, it is unclear if per spot only one detection experiment can be performed? Could lower DNA amounts be traded off with longer detection times?

Response: We thank the reviewer for the enlightening comment and inspiring suggestions. Indeed, the amount of DNA required is one of the topics we are working on for future improvements. The amount of DNA used in the detection experiment is for single use (the reading process is a destructive process for DNATags). We have added the following paragraphs talking about this limitation, our currently suitable application given this limitation, and our future directions to improve this issue in 'Discussion' section:

"Although the current cost of DNATag remains high (\$2.77 for a 24-bit DNATag), which may present challenges for adoption in low-margin markets like agriculture, a promising entry point for DNATag is in enhancing the anti-counterfeiting measures of luxury goods—such as wines, paintings, and jewelry—that are particularly vulnerable to counterfeiting due to their high value and profit margins. These products often rely on existing anti-counterfeiting technologies, such as RFID, NFC chips and hologram tags. DNATags, with their molecular-level security features, offer a unique layer of protection that could further safeguard these high-value items."

Despite its advantages, DNATrack has limitations. One major challenge is the significant DNA quantity required for a single readout ($\sim 60 \mu\text{g}$ per destructive readout), primarily due to the large reaction buffer ($60 \mu\text{L}$) needed to cover the DNATag-reading ticket ($\sim 1 \text{ cm}^2$). Since

reaction speed is semi-proportional to DNA concentration, the large buffer necessitates higher DNA inputs to ensure efficient detection. To address this, smaller DNATag-reading tickets with reduced size of reporter spots can be produced using a microarray printer, or the reaction time can be extended, as signal intensity is semi-proportional to both DNA concentration and reaction time.“

c) Forgeability:

The authors write, that the novel tracers can not be forged, however it is unclear why the following attack is not possible:

- 1. Extract DNA from product*
- 2. split into two fractions*
- 3. Sequence the first fraction to get all the sequences present in the taggant*
- 4. Synthesise reporter oligos to all sequences identified, and create a "self-made" ticket, according to the description in the manuscript*
- 5. Test the second fraction of DNA on the "self-made" ticket to identify which of the sequences present in the taggant are blocked to reveal the codeword*

It is evident that this approach is much more elaborate than forging a standard DNA barcode, but it does not seem unforgeable, and also comes with significant disadvantages compared to a standard DNA barcode (not amplifiable).

Response: We thank the reviewer for proposing such an intriguing potential. This attack can be mitigated using several approaches, which we have now included in the manuscript. For the reviewer’s convenience, we provide the relevant text below.

“A more sophisticated hacker could perform an advanced read-and-rewrite attack. In this scenario, the DNA is extracted from the product and split into two fractions. The first fraction is sequenced to identify all sequences in the taggant, and reporter oligonucleotides are synthesized to create a "self-made" ticket. This ticket is then used to test the second fraction of DNA, identifying which sequences in the taggant are blocked and thereby revealing the codeword.

This attack can be mitigated using several approaches. First, DNA Bits can be chemically modified at their ends to render them incompatible with sequencing, as sequencing typically requires ligation of DNA to sequencing adapters. Another approach is to introduce redundant DNA sequences that share the same structural format as DNA Bits (i.e., T*U15*S22*). A large pool of random redundant sequences can be inexpensively generated by incorporating random mixed bases in the Si domain (e.g., NNNN..., where N represents a mixture of A/T/C/G). This strategy yields 4^{20} possible sequences for the 20-nt Si domain, making it prohibitively expensive to reverse-engineer the authentic sequence from this vast set of random sequences.”

We acknowledge that it is inherently challenging to create a tagging scheme that is entirely forgery-resistant against a determined and capable adversary. That said, we believe our tagging scheme significantly raises the barrier to counterfeiting, as the cost of compromising our DNATags far exceeds the cost of producing them. This cost asymmetry provides

substantial value for practical applications. Moreover, high-value products typically employ multiple anti-counterfeiting technologies, such as QR codes, watermarks, NFC tags or holograms. As a molecular technology, DNATags can be seamlessly integrated with these existing methods to enhance overall security.

Recognizing that security warrants further investigation in the future, we have expanded our discussion on this topic in the 'Discussion' section:

“Lastly, while DNATags improve security, further investigation is needed to address potential forgery attacks. With the increasing accessibility of DNA amplification and sequencing, identifying and mitigating vulnerabilities is crucial to ensure DNATrack’s long-term effectiveness.”

In short, the method is not as non-forgable as described, the stability of the taggants for over a week is unclear, and very high amounts of taggants are required per spot. While the idea is innovative, and an improvement over reference 3, the manuscript neither goes into the physio-chemical performance of the original idea, nor is the application level data shown convincing (and much thereof is hidden in the Supp. Info).

Response: We thank the reviewer for their careful thoughts and constructive comments. We have included responses to these points (please see above). We appreciate all the improvements brought to our work by the reviewer’s advice.

Also, the manuscript has a structural problem, as the last page of the introduction is rather a conclusion, which comprises some greatly overstated aspects of the technology, which are not supported by the data and or describe the work performed. (i.e. protection from amplification attacks, "significantly improved stability", "we....systematically assess reaction latency and error rates"). This section of the manuscript requires shortening, and the absence of any performance and conclusion statements (as these should follow the experiments and discussion). This section should rather be used to pose a hypothesis, which is then proven or rejected in the following sections of the manuscript.

Response: We thank the reviewer for pointing out the structural issue of our writing. We have replaced the last 4 paragraphs in the ‘Introduction’ section with the following 2 paragraphs:

“We present DNATrack, a prototype supply chain tracking system featuring DNA tags, aiming at addressing the existing design gap (Fig. 1). DNATrack employs specially designed DNA mixtures, termed DNATags, that store digital information and label products for tracking. We purposefully designed the DNATag to make them inexpensive to manufacture at scale. DNATags incorporate a unique DNA hybridization encoding, termed HyEn, that provides protection against PCR forgery or read-and-rewrite attacks, significantly heightening the tag security compared to existing DNA tagging schemes. We utilized a paper-based readout, termed DNATag reading ticket, or ticket, that translates DNATags into visual signals representing the stored digital information (Fig. 1} (b)). Users can conveniently read DNATags in the field within 15 minutes at a cost of ~\$2-\$4 per read, requiring only basic equipment: an app-equipped smartphone and a fluorescence reader. Our work showcases a full end-to-end workflow, encompassing

information encoding methods (both digital and physical), object labeling, to the final field-deployable readout.

In our work, we demonstrated that DNATags reliably induce distinct and correct fluorescence signals on the DNATag reading tickets. We assessed the reaction latency and accuracy. For consistency of the readout, we integrated automated image processing and an error correction code (ECC) in silico. To ensure reliability and reproducibility, we experimented with varied solvent conditions of the DNATags and found out the optimal setup for best stability. We validated DNATrack end-to-end by labeling different objects including lettuce, corks, and petri-dishes with 24-bit DNATags carrying randomly generated barcodes, storing the objects overnight, and reading out the DNATags. Finally, we highlighted the anti-forgery feature of our HyEn encoding scheme by showcasing failed PCR-based forgery against DNATags next to successful forgery attacks of other existing DNA tagging schemes with a simple protocol.”

More minor details:

The data quality of FIG 2f (far right) is very low. Is this from only one experiment, and why is the fluorescence decay shown on a log10 scale?

Response: We thank the review for pointing this out. Indeed, the data quality in Figure 2 (f) was not on par with other data because this experiment was done at our initial stage, and we were not proficient with our experimental workflows and image capturing skill. We have redone the experiment with triplicates to ensure robustness and reproducibility and updated Figure 2 (f) with new data. We are attaching a snapshot of updated Figure 2 here for convenience of reading:

In terms of the scale, we plotted the fluorescence decay in log10 scale for easier reading. We now have included the plots in linear scale in the supplementary materials (Figure S4). We are attaching a snapshot here for convenience:

Fig 3c: The fit to the normalised data for the 0 bit is very poor. A better description is required (and might be found in bivariate normal distributions with Fisher transformation)

Response: We thank the reviewer for pointing this out. We have revised our fitting and we believe the truncated Gaussian distribution should be a more appropriate model since our data is hard cut at 0 and 1. Therefore, we redid the fitting and described the method in the ‘Gaussian fitting of normalized signal decay’ subsection under ‘Method’ section, as quoted below for convenience. We have also updated Figure 3 (c) and we are attaching a snapshot here for convenience:

“Gaussian fitting of normalized signal decays. We gathered the normalized fluorescence decays of all reporter spots from all our well-controlled experiments (i.e., excluding those experiments aimed at testing varied experimental conditions) and separated the data into two sets by whether they encountered corresponding Bit ‘1’ or ‘0’. We directly fitted the dataset of Bit ‘1’ using truncated Gaussian distribution, setting the truncation boundaries as ‘0’ and ‘1’. For the Bit ‘0’ dataset, since the expected mean value is 0 because there should be no reaction, while the actual data were all larger than 0, we created a dummy dataset by pooling together the Bit ‘0’ dataset and their opposite values and fitted this dummy dataset with Gaussian distribution.”

The experimental description is short, and the reporter complex (dye and quencher) are not described. Also, the quenchers are not mentioned in the sequence listing. Neither is the application and removal from the cork and salad not described. More specification on "orange filter window" is required. Also, the experimental section to the device presented in Figure S11 is missing.

Response: We thank the reviewer for pointing out the issues. We have added the following paragraph describing the molecular design of the reporter complex (which do not include a quencher by our design) in ‘Molecular design and reaction assessment’ section:

“Reading DNATags utilizes toehold mediated strand displacement on a nitrocellulose paper ticket, producing a fluorescent pattern [2]. A set of fluorophore-labeled DNA reporter complexes containing different Si domains are arrayed on a nitrocellulose paper ticket. Each reporter complex consists of a probe strand $\langle X S_j UT \rangle$, an anchor strand $\langle U^*S_j^* \rangle$, and a fluorophore strand $\langle FX^* \rangle$ (Fig. 2 (C)). The anchor strand contains a 30-nt poly-T tail on its 3’ end for enhanced binding to the nitrocellulose paper (Fig. S3). Upon contact, DNA Bit 1 strand $\langle T^*U^*S_i^* \rangle$ can bind the toehold domain of the probe strand and subsequently displace the probe strand together with the fluorophore strand off the anchor strand if the S_i^* matches the S_j domain (i.e. $i = j$). The displaced fluorophore strands can be easily washed away, resulting in significantly reduced fluorescent signals. In contrast, the ‘0’s of DNA Bits cannot displace the probe strand because the toehold region $\langle T \rangle$ is covered by a blocker strand, resulting in unchanged high fluorescence. Thus, the DNATag reading ticket initially displays an array of uniformly bright fluorescent dots, and it switches to a specific pattern of bright and dim spots upon exposure to a DNATag.”

We have added the following subsection in ‘Methods’ section to describe the experimental workflow of application and removal of the DNATags from the cork and lettuce:

“Labeling objects with DNATags and retrieving the DNATags for readout. We labeled the corks, lettuce, or petri-dishes with DNATags by dropping the 60 μL DNATag solution onto the surface of the object and letting it dry. The labeled object was kept at room temperature overnight before readout if not specified otherwise. To retrieve the DNATag, we dropped 60 μL water onto the dried DNATag, pipetted up and down several times to dissolve the DNA, and move the solution to a tube as the retrieved DNATag solution. The solution can then be applied to the DNATag reading ticket for readout.”

We have added item specifications of the orange filter that we used for the home-built device in Figure S11 caption:

“The device consists of 20 blue LEDs (12V input, 5 LEDs on each side of the ticket), one orange filter (Tiffen 49mm 21 Filter purchased from Amazon), one 12V power supply (for powering all 20 LEDs), and some 3D printed frame structure. The total cost of all components is \$50.”

We did not conduct any DNATag reading experiment with the home-built device, nor did we evaluate the performance of the device compared to the gel reader. Therefore, we did not have a subsection related to it in the ‘Method’ section. We built this device as a proof of concept, so all we did was placing a ticket from previous experiment and briefly demonstrated its capability of displaying the fluorescence signal correctly. We have added more details of the LED specification, filter, and power supply to Figure S11 caption.

some typos: "This has" instead of "This have" (second sentence); "carring" instead of "carrying"; "Ref [?, 23, 33]", one reference=?

Response: We thank the reviewer for careful reading. We have carefully corrected typos, grammar mistakes, and references.

Author's Response to Reviews of:

Hybridization-Encoded DNA Tags with Paper-Based Readout for Anti-Forgery Raw Material Tracking

Response to Reviewer #1:

I am generally satisfied with the additional data provided and the text changes made in the manuscript. Since readers may have similar questions to mine, please add text to briefly describe the -20°C storage results (Figure S9) in the main manuscript, and also include some discussions of the previous questions 4 and 6 (natural DNA interference and readout turnaround time).

We thank the reviewer for their helpful suggestions and their approval of our added data and revisions. We have made the remaining changes according to their suggestions, including:

- **We added the following text describing our results in Figure S9 to the 'Hybridization Encoding and reaction assessment' section:**

"The T* domain serves as a toehold for initiating strand displacement reaction during the readout, while the U* domain provides stability of the partial double-strand structure at the intended working temperature. According to our simulation and experimental test, a U* domain with the length of 15 nucleotides can ensure the stability of DNA Bits at room temperature or below including frozen (Fig. S9). For applications requiring higher ambient temperatures, the U* domain can be elongated to increase the melting temperature (T_m) (Fig. S16)."

- **We added the following sentence in the 'Hybridization Encoding and reaction assessment' section:**

"Reading DNATags utilizes toehold mediated strand displacement on a nitrocellulose paper ticket, producing a fluorescent pattern [2, 42, 43]. This reaction is highly efficient and specific so that the reading can be carried out in a few minutes and natural nucleic acids will not interfere with the reaction."

- **We added the following sentence in the 'Discussion' section:**

"Meanwhile, faster turnaround time will also be our future goal. We plan to achieve it by utilizing multiple sets of toehold domains T to minimize the kinetic crosstalk between different DNA Bits and selecting an optimal fluorophore that yields a higher signal-to-noise ratio."

Response to Reviewer #2:

The authors improved manuscript based on my review report and all my concerns have been addressed. Therefore, I recommend this manuscript for publication.

Response: We thank the reviewer for their careful readthrough and their approval of our revisions.

Response to Reviewer #3:

The authors have significantly reworked the manuscript and have taken this reviewers comments into account.

While I still find the explanation of the scheme complicated to comprehend, this might be unavoidable (due to the nature of cryptographic schemes). I propose publishing the manuscript in the present form.

Response: We thank the reviewer for their careful readthrough and their approval of our revisions and added results.

Response to Reviewer #4:

Response: We thank the reviewer for their efforts reading through our manuscript.